# Towards Efficient Gradient-Based Meta-Learning in Heterogenous Environments

## Abstract

A challenging problem for machine learning is few-shot learning, as traditionally, models trained with SGD require many training samples to converge. Since meta-learning models have strong fine-tuning capabilities for the distribution of tasks, many of them have been applied to few-shot learning. Model-agnostic meta-learning (MAML) is one of the most popular ones. Recent studies showed that MAML-trained models tend to reuse learned features and do not perform strong adaption, especially in the earlier layers. This paper presents an in-detail analysis of this phenomenon by analyzing MAML's components for different variants. Our results show an interesting relationship between the importance of fine-tuning earlier layers and the difference in the distribution between training and testing. As a result, we determine a fundamental weakness of existing MAML variants when the task distribution is heterogeneous, e.g., the numbers of classes or the domain do not match during testing and training. We propose a novel nonparametric version of MAML that overcomes these issues while still being able to perform cross-domain adaption.

## 1 Introduction

Learning tasks from only a few observations is known as *few-shot learning* and of major interest in the machine learning community (Finn et al., 2017; Vinyals et al., 2016; Snell et al., 2017; Cai & Shen, 2020; Tseng et al., 2020). Usually, a problem is solved by minimizing the empirical risk with many training samples in many iterations. However, humans learn new tasks very quickly by using knowledge obtained in their lives a priori (Salakhutdinov et al., 2012). Meta-learning is motivated by how humans learn, where the goal is learning *how to learn* and a common approach to solve few-shot learning problems due to its ability to efficiently leverage information from many tasks.

*Model-Agnostic Meta-Learning* (MAML) has been one of the most successful meta-learning algorithms for few-shot learning in recent years (Finn et al., 2017). In MAML, the network is meta-optimized for fast gradient-descent based fine-tuning on an unseen task. Its formulation of the meta-learning objective inspired a plethora of research (Yoon et al., 2018; Li et al., 2017; Vuorio et al., 2019; Finn et al., 2018), to the extent that MAML exists both as a concrete meta-learning algorithm but also as a paradigm that influences meta-learning methods to this day.

Previous work has discussed whether MAML actually allows rapid fine-tuning or simply leverages its meta-representations effectively (called *feature reuse*). Raghu et al. (2020) found out that freezing the earlier layers of a network during fine-tuning improves the performance, meaning that fine-tuning of the network body is not the major factor contributing to its few-shot capabilities, indicating feature reuse. Oh et al. (2021) discovered that in the case of cross-domain adaptation, a change in the earlier layers is beneficial and proposed to fix the network head instead to enforce earlier weight change, a method they call *body only inner loop* (BOIL). However, as we will argue in Section 3, its fixed final layer is impractical when the numbers of classes differ across tasks, which is a considerable limitation in real-world scenarios.

In this paper, we develop a novel technique called NP-MAML, which has a nonparametric head but is still trainable via gradients. Similar to BOIL, NP-MAML enforces changes in earlier layers to solve cross-domain tasks. In addition, it is flexible to the heterogeneous task distribution. We

compare the performance and representation change of these approaches under different challenges: cross-domain adaption and task dimensionality. We further deliver an analysis of the different components of the network and want to specify their respective role with regard to fine-tuning and task adaptation.

## 2 META-LEARNING FOR FEW-SHOT LEARNING

In few-shot learning, a dataset consists of tasks coming from a task distribution. Each task contains of a support set $S$ with labels samples and a query set $Q$ where predictions have to be made on the samples (Vinyals et al., 2016). A typical support set consists of $K$ examples for each of $N$ classes, wherefore a problem is usually described as $N$-*way*-$K$-*shot* classification problem.

Next to non-episodic approaches (Gidaris & Komodakis, 2018; Qi et al., 2018; Chen et al., 2019), many meta-learning methods have been applied to few-shot learning problems (Vinyals et al., 2016; Li et al., 2017; Yoon et al., 2018; Snell et al., 2017). They are particularly useful as they can learn a configuration from which it is easy to solve various tasks. The configuration has seen similar features or recurring patterns shared across the tasks and allowed to be transferred to novel unseen tasks.

### 2.1 OPTIMIZATION-BASED META-LEARNING

Optimization-based meta-learning (Ravi & Larochelle, 2017) follows the idea of meta-learning a task-specific optimizer $U$, that transforms some initial set of parameters $\theta$ into task-parameters $\phi$. Although $U$ can be chosen arbitrarily, it is typically modelled as a $m$-step, gradient-based update scheme, denoted $U^{(m)}(\theta)$. While methods like Meta-LSTMs model $U^{(m)}(\theta)$ explicitly via an LSTM (Ravi & Larochelle, 2017), which iteratively transforms $\theta$, given both loss and loss gradient, the popularity and success of MAML is due to its efficient and model-agnostic design, allowing $U^{(m)}$ to take any differentiable form and rather meta-optimizing $\theta$, the initial parameters, leading to superior performance and more flexibility.

### 2.2 MODEL-AGNOSTIC META-LEARNING

MAML (Finn et al., 2017) introduces a couple of new interpretations of the meta-learning problem that allow its efficient and flexible training. Firstly, task data is assumed to come from a *task distribution* $p(\tau)$, which in turn will allow us to form an expectation over task performance. Secondly, let $L_\tau(\mathbf{x}; \theta^*)$ denote the loss of a model $f_\theta$ on task-data $\mathbf{x} = \{(x_1, y_1), ..., (x_T, y_T)\}$, parameterized by $\theta^*$. To stay consistent with the notation introduced hitherto, we will denote with $S_\tau$ the support set of task $\tau$ and with $Q_\tau$ the query set of task $\tau$. Then, we can express the optimization objective of MAML as

$$\min_\theta \mathbb{E}_{\tau \sim p(\tau)}\Big[ L_\tau(Q_\tau; U^{(m)}(S_\tau; \theta)) \Big], \tag{1}$$

where we write $U^{(m)}(S_\tau; \theta)$ to denote a $m$-step optimizer, transforming meta-parameters $\theta$ given the support set $S_\tau$. Intuitively, MAML improves on-task performance on average by optimizing the initial parameters of model $f_\theta$ and subsequently fine-tuning those parameters on the support set $S_\tau$, where on-task performance is measured by evaluating the fine-tuned model on the query set $Q_\tau$.

As $p(\tau)$ is high-dimensional and typically unknown, computing the actual expectation integral is not feasible, which is why we define the meta-loss of MAML as

$$\mathcal{L}(\theta) = \frac{1}{|\mathcal{T}|} \sum_{\tau \in \mathcal{T}} L_\tau(Q_\tau; U^{(m)}(S_\tau; \theta)), \tag{2}$$

where $\mathcal{T}$ is a batch of tasks, sampled from $p(\tau)$, and where we replace the expectation with an empirical mean. This meta-loss is then optimized with standard gradient descent with step-size $\beta$, i.e.,

$$\theta^{(t)} = \theta^{(t-1)} - \beta \nabla_{\theta^{(t-1)}} \mathcal{L}(\theta^{(t-1)}). \tag{3}$$

A crucial detail of MAML is that the gradient descent update in Equation 3 involves second-order derivatives if task optimizer $U^{(m)}$ is gradient-based. The authors thus propose a first-order approximation to the update that performs just as well as the full-gradient version of MAML. Other approximations are proposed by (Nichol et al., 2018; Rajeswaran et al., 2019). Other extensions of MAML stabilized and improved the method by proposing various modifications (Antoniou et al., 2019) decoupling the gradient-based adaption procedure (Rusu et al., 2019) and integrating target samples into the fine-tuning step (Antoniou & Storkey, 2019).

### 2.2.1 RAPID LEARNING OR FEATURE REUSE

After investigating the adaptation behavior of MAML, Raghu et al. (2020) claimed that MAML tends to learn nearly fixed representations in the network body rather than rapidly fine-tuning them, a phenomenon they call *feature reuse*. They analyzed the representation similarity of the layers before and after fine-tuning and observed hardly any changes in the early layers. Their proposed technique ANIL (almost no inner loop), which freezes the network body during fine-tuning, slightly outperforms MAML on several tasks like MiniImageNet and Omniglot and is additionally faster. Oh et al. (2021) proposed BOIL (body only inner loop), a complementary approach freezing only the network head during fine-tuning, showing that in the case of (cross-)domain adaption, a fast adaptation of weights in the earlier layers, especially the penultimate layer, is not only possible but also highly beneficial to performance.

### 2.2.2 MODULARIZATION OF MAML

To accumulate the perspective of previous research and ours, we formulate a *modular interpretation* of MAML. Raghu et al. (2020) and Oh et al. (2021) have found out that fine-tuning different components of the architecture affect the runtime and the overall performance. They analyze the effects of MAML by performing experiments on few-shot image datasets such as MiniImageNet with an architecture consisting of four convolutional blocks and a linear layer, which we further call *conv4*. We will later discuss each component of the conv4 architecture in the context of the algorithmic and architectural challenges.

Let $f_\theta$ be a model parameterized by $\theta$. We modularize $f_\theta$ by partitioning the $f_\theta$ into sub-components, e.g., a single layer, a group of layers that form one modular unit, or an entire sub-network. We call $\mathbf{g} = \{g^{(1)}(\cdot, \theta_1), ..., g^{(n)}(\cdot, \theta_n)\}$ a modularization of size $n$ of $f_\theta$ if and only if

$$f_\theta = g^{(n)}(\cdot, \theta_n) \circ ... \circ g^{(1)}(\cdot, \theta_1) \tag{4}$$

and

$$\theta = \theta_n \cup ... \cup \theta_1. \tag{5}$$

Furthermore, we denote with $\mathbf{g}^* \subseteq \mathbf{g}$ the components that receive task-specific fine-tuning and call $\mathbf{g}^*$ a *meta-learning configuration*. Then, the meta-update according to MAML for parameter $\theta_i$ becomes

$$\theta_i = \theta_i - \beta \cdot \nabla_{\theta_i} L(Q; \phi_i), \tag{6}$$

with

$$\phi_i = \begin{cases} U^{(m)}(S; \theta_i), & \text{if } g^{(i)} \in \mathbf{g}^* \\ \theta_i, & \text{otherwise,} \end{cases} \tag{7}$$

where $S$ is the support set and $U^{(m)}$ is an optimizer with $m$ steps. We can recover the original MAML by setting $\mathbf{g}^* = \mathbf{g}$.

We study the advantages and challenges of different configurations of $\mathbf{g}^*$. We partition the conv4 network into three components: $g_{\mathbf{early}}$ denotes the first three convolutional blocks, $g_{\mathbf{penult}}$ denotes the last convolutional block (penultimate layer), and $g_{\mathbf{head}}$ denotes the last linear layer. Previously proposed methods can be obtained by only optimizing the weights of selected sub-components (see Table 3 in Appendix A).

## 3 Challenges of Few-Shot Learning

Standard few-shot learning benchmark datasets like Omniglot or MiniImageNet are single-modal, homogenous, and balanced, whereas real-world datasets are multi-modal, heterogeneous, and imbalanced (Vuorio et al., 2019; Triantafillou et al., 2020). The data can come from different domains (multi-modal) and have a flexible number of shots (imbalanced) and ways (heterogeneous). We divide the challenges into two groups, distinguishing between *algorithmic* and *architectural* challenges[1] and discuss how the methods from Section 2.2.1 may deal with these challenges.

### 3.1 Algorithmic Challenge: Domain Shift

*Domain shift* is a straightforward yet challenging problem of cross-domain adaptation (Oh et al., 2021; Triantafillou et al., 2020; Cai & Shen, 2020; Tseng et al., 2020). Therein, a model is meta-trained with a task from a domain $A$ and is meta-tested with a task from domain $B$, which has not been available during training. Oh et al. (2021) showed that BOIL outperforms MAML and ANIL in this setup, as they learn the general features for domain $A$ and are not able to adapt these features for the different domain $B$, even when domain shift necessitates adaptation. Thus, our proposed technique in Section 4 aims at maintaining the cross-domain adaptation capabilities of BOIL. A more detailed discussion on domain adaption can be found in Appendix B.

### 3.2 Architectural Challenge: Task Dimensionality

Under *task dimensionality*, we summarize the dimensionality of both the samples and the labels within a task. Whereas it is always possible to scale images to a common $n \times m$ pixel grid, for the output dimensionality of a task (ways), there is no trivial modification besides re-initialization. This limitation is caused by the fully-connected linear layer whose weights need a fixed output size to be transferred[2]. The head of the network turning a hidden representation $g_{\mathbf{body}}(x; \theta)$ into logits and probabilities is typically expressed as

$$p(\hat{y} \mid x) = \sigma\Big(W^T \cdot g_{\mathbf{body}}(x; \theta) + b\Big), \tag{8}$$

where $x$ is an input sample, parameters $W \in \mathbb{R}^{H \times N}$ and $b \in \mathbb{R}^N$ form the linear layer, $H$ is the output dimension of the network body and $\sigma$ is the softmax operation. Since the dimension of $W$ has to be fixed, a linear architecture is limited because it cannot adapt distributions with varying output dimensions. A parametric solution of this is proposed by Triantafillou et al. (2020) where the prototypes of the support set form the initialization for the final linear layer. Another proposed solution is UnicornMAML, where the same weight is meta-learned for every neuron in the output layer and replicated in the inner loop to match the task dimensionality Ye & Chao (2022). However, both approaches explicitly construct a head to fit the parametric paradigm. The nonparametric version we propose in the following section recovers predictions implicitly from support set representations and alleviates the need for a physical linear layer altogether.

## 4 Nonparametric Model-Agnostic Meta-Learning

In this section, we propose a novel technique called nonparametric model-agnostic meta-learning (NP-MAML), which can deal with both algorithmic and architectural challenges of Section 3. In addition, it also is implicitly permutation invariant, a quality standard MAML approaches lack, as discovered by Ye & Chao (2022).

We derive our method as follows. Raghu et al. (2020) proposed a nonparametric head for MAML and suggested no updates in the inner loop (NIL := No Inner Loop ) during fine-tuning. Utilizing metric-based approaches of Vinyals et al. (2016) and Snell et al. (2017) in NIL, a similarity metric

---

[1]architectural challenges take the form of simple, physical limitations like mismatches of in- and output dimensions and algorithmic refers to problems with performance, lack of generalization or instability of the results (such as very high variance on the classification accuracy)

[2]Although we will not study the effect of rescaling in this work, we encourage a further investigation into whether rescaling has an impact on the predictive performance of meta-learning models for image processing.

$d$ is applied to the representation $g_{\mathbf{body}}(x)$ and the prototype $c_i$, defined as

$$c_i := \frac{1}{K} \sum_{x \in S_i} g_{\mathbf{body}}(x), \tag{9}$$

for every class label $i$ of the support set $S$

$$p(\hat{y} \mid x, S) = \sum_{i=1}^{N} k(-d(g_{\mathbf{body}}(x), c_i)) \cdot y_i, \tag{10}$$

with $k(\cdot)$ as a kernel normalizing the distance measures into a probability and where again $g_{\mathbf{body}}(x)$ denotes the representation of samples $x$ by the network body. Coincidentally, this approach is very adaptive to the task dimensionality during testing, as predictions are formed on the dimensionality of the body representations. The NIL had is explicitly designed to leverage the strong meta-representations acquired during MAML training, demonstrating that the network still performs well on in-domain tasks, even when no test-time fine-tuning is performed. However, it is well known in the meta-learning community that models generally benefit from an agreement of train- and test-time loss Vinyals et al. (2016). To enable such conditions for a NIL-type predictor, we propose a new method, nonparametric model-agnostic meta-learning (NP-MAML), which replaces the parametric head of MAML with a nonparametric head also for training. It allows updating the earlier layers during fine-tuning and can be incorporated during meta-optimization. Since the nonparametric head does not require a fixed output size, it can deal with architectural challenges. At the same time, head predictions are formed entirely based on the body representations, enforcing a BOIL-type fine-tuning of the network body that is beneficial to the cross-domain setting.

We note that applying a NIL head at test time to a model trained with BOIL is synthetically possible but leads to non-meaningful results, as the model still relies on the representations learned in the head. We verify this claim experimentally in Section 5.2, where we use NIL-testing on BOIL to architecturally cope with variations in the number of ways at test time.

In NP-MAML, we use the predictive distribution of Equation 10 and extend it such that it allows gradients to flow through the fine-tuning stage:

$$p(\hat{y} \mid x, S) = \sum_{i=1}^{N} k(-d(g_{\mathbf{body}}(x; U(S; \theta)), c_i)) \cdot y_i, \tag{11}$$

where $d$ is again a similarity metric, measuring the similarity between the representation $g_{\mathbf{body}}(x; U(S; \theta))$ of sample $x$ by the network body and prototype $c_i$. Note that in contrast to NIL, the parameters of the body representation are the result of fine-tuning with $U$, a special one-step optimizer that is similar to a contrastive loss (Andonian et al., 2021), defined as

$$U(S; \theta) = \theta - \frac{\alpha}{K} \nabla_\theta \Big( \sum_{i,j} -d(g_{\mathbf{body}}(S_i; \theta), c_j) \Big). \tag{12}$$

Intuitively, $U$ fine-tunes the network such as to maximize the extra-class distances between representations of support set samples, meaning that features from different classes should be less similar than features from the same class. Here, we use the same distance metric $d$ from which we also form predictions on the query set in Equation 11. We further visualize the fine-tuning effect on the network's features in Appendix C, where we see that the fine-tuning step acts as a pulling force on the metric space, disentangling intra-class from extra-class features. We hypothesize that NP-MAML fine-tuning learns to condition the metric space, such as to ease classification.

Note that although the prototypes $c_j$ also depend on $\theta$, we stop the gradient flow for the prototype when calculating the meta-gradient, achieving similar runtime performance as the other methods. Although this is not a first-order approximation of the meta-gradient, this simplification is based on previous evidence that for successful meta-learning, not the entire meta-gradient is required (Finn et al., 2017).

## 5 EXPERIMENTS

In this section, we show the results of several experiments, which empirically support that NP-MAML can deal with heterogeneous challenges while ensuring cross-domain adaption and outperforming previous adaptations of MAML. In addition, we analyze the different components of

**In- and Cross-Domain Classification**

| | MiniImageNet | | FC100 | |
|---|---|---|---|---|
| | 1-shot | 5-shot | 1-shot | 5-shot |
| ANIL | $47.69 \pm 0.62$ | $62.58 \pm 0.54$ | $32.83 \pm 0.53$ | $43.20 \pm 0.53$ |
| BOIL | $\mathbf{50.16 \pm 0.64}$ | $65.31 \pm 0.53$ | $36.03 \pm 0.57$ | $47.83 \pm 0.52$ |
| MAML | $45.42 \pm 0.61$ | $61.84 \pm 0.55$ | $34.23 \pm 0.55$ | $44.50 \pm 0.54$ |
| NP-MAML | $49.82 \pm 0.64$ | $\mathbf{67.39 \pm 0.53}$ | $\mathbf{38.57 \pm 0.57}$ | $\mathbf{50.86 \pm 0.54}$ |
| Proto-Net | $48.12 \pm 0.81$ | $66.17 \pm 0.67$ | $31.41 \pm 0.67$ | $42.78 \pm 0.70$ |
| UnicornMAML | $48.74 \pm 0.87$ | $63.44 \pm 0.76$ | $35.95 \pm 0.80$ | $45.15 \pm 0.76$ |

Table 1: In this table, the **classification accuracy [%]** of all models on a test set of MiniImageNet and FC100 are depicted. All models have been trained on MiniImageNet.

MAML and show how each part of the network contributes to performance and representation change. All models are trained in a *homogeneous* environment and are subsequently tested in homogeneous and heterogeneous environments. As a first step, this allows for more fine-grained control of cause and effect relationships with regard to predictive performance than we would have if we applied it on Meta-Dataset Triantafillou et al. (2020). Our setting is also most realistic for real-world applications, where we meet environmental conditions unexpected at training time. Note, for example, that in our setting, models are not explicitly *trained* to cope with domain shift, yet showing strong performance in unexpected environment changes is a crucial property in real-world problems. Despite all this, we regard a validation of our findings on Meta-Dataset as an important further step.

We design the experiments in a way that the effects of individual parts of the environment (domain, number of ways, and shots) are investigated and partially isolated. We train the conv4 architecture, the standard architecture in previous literature on few-shot learning with MAML (Finn et al., 2017; Raghu et al., 2020; Oh et al., 2021; Vuorio et al., 2019; Yoon et al., 2018; Finn et al., 2018) on MiniImageNet to solve a $N$-way-$K$-shot classification task. We further model domain shift by testing on a variety of few-shot learning datasets, such as FC100 (Oreshkin et al., 2018), CIFAR-FS (Devos et al., 2019) and CUB (Wah et al., 2011), and measure the test accuracy. We further provide test-time accuracy for MiniImageNet to study the impact of NP-MAML when going from the in- to the cross-domain setting. We choose the hyper-parameters of the original MAML-approaches (Finn et al., 2017; Raghu et al., 2020; Oh et al., 2021) summarized in Appendix D. For all performance results, we show classification accuracy on a hold-out set, where the $\pm$ indicates the $95\%$ confidence interval over tasks, following Finn et al. (2017). We further average these results across multiple seeds. We provide results for MiniImageNet and FC100 in the main body of the text and refer to Appendix E for results on CIFAR-FS and CUB, as well as to Appendix F for studying an increase in the test-time environment change also with respect to the number shots and to Appendix G for a detailed analysis of the importance of very early network layers on in- and cross-domain problems.

## 5.1 DOMAIN SHIFT

In our first experiment, we train each model on 5-way-1-shot and 5-way-5-shot on MiniImageNet and test it on MiniImageNet (in-domain) and FC100 (cross-domain), where the numbers of shots $K$ matches its training conditions.

In Table 1, we observe that our approach NP-MAML is competitive with BOIL on 1-shot MiniImageNet and better in all other cases. As expected, ANIL performs especially poorly in the cross-domain setup, as an adaptation of the body representations is crucial to cope with domain shift.

## 5.2 META-FEATURE QUALITY WHEN VARYING NUMBER OF WAYS

In this experiment, we look at the quality of the features of the penultimate layer, which we call *meta-features*. The model is trained as in Section 5.1 on MiniImageNet on the homogeneous 5-way-$K$-shot problems. We test the model on both MiniImageNet and FC100 but vary the number of ways of each task ($N = 4, 7$ and $10$). Since MAML, ANIL and BOIL can architecturally not fine-tune the last layer (see Section 4), we use the NIL-testing proposed by Raghu et al. (2020). NP-MAML can be fine-tuned, and we depict both results without fine-tuning (0 inner loop steps) and with fine-tuning NP-MAML (1 inner loop step).

**(a) 1-shot mixed-ways**

| | MiniImageNet | | | | FC100 | | | |
| --- | --- | --- | --- | --- | --- | --- | --- | --- |
| | 4-way | 5-way | 7-way | 10-way | 4-way | 5-way | 7-way | 10-way |
| ANIL | $55.29 \pm 0.77$ | $49.64 \pm 0.63$ | $40.78 \pm 0.49$ | $32.96 \pm 0.36$ | $39.03 \pm 0.67$ | $33.61 \pm 0.53$ | $26.39 \pm 0.42$ | $20.62 \pm 0.31$ |
| BOIL | $30.98 \pm 0.45$ | $25.90 \pm 0.38$ | $19.17 \pm 0.28$ | $14.47 \pm 0.21$ | $29.68 \pm 0.45$ | $24.50 \pm 0.36$ | $18.31 \pm 0.28$ | $13.48 \pm 0.21$ |
| MAML | $52.60 \pm 0.73$ | $47.11 \pm 0.63$ | $38.61 \pm 0.47$ | $30.79 \pm 0.34$ | $40.84 \pm 0.69$ | $35.09 \pm 0.57$ | $27.22 \pm 0.44$ | $21.23 \pm 0.30$ |
| NP-MAML (0) | $49.18 \pm 0.70$ | $43.44 \pm 0.60$ | $35.20 \pm 0.44$ | $28.38 \pm 0.34$ | $40.24 \pm 0.64$ | $34.76 \pm 0.52$ | $27.38 \pm 0.40$ | $21.24 \pm 0.31$ |
| NP-MAML (1) | $\mathbf{55.50 \pm 0.75}$ | $\mathbf{49.82 \pm 0.64}$ | $\mathbf{41.48 \pm 0.47}$ | $\mathbf{33.59 \pm 0.36}$ | $\mathbf{44.09 \pm 0.67}$ | $\mathbf{38.57 \pm 0.57}$ | $\mathbf{30.37 \pm 0.43}$ | $\mathbf{23.83 \pm 0.30}$ |
| ProtoNet | $53.91 \pm 0.95$ | $48.12 \pm 0.81$ | $40.02 \pm 0.61$ | $32.36 \pm 0.46$ | $36.56 \pm 0.78$ | $31.41 \pm 0.67$ | $24.54 \pm 0.51$ | $18.99 \pm 0.37$ |
| UnicornMAML | $53.18 \pm 0.99$ | $48.74 \pm 0.87$ | $39.87 \pm 0.68$ | $31.88 \pm 0.49$ | $41.78 \pm 0.94$ | $35.95 \pm 0.80$ | $27.78 \pm 0.61$ | $21.87 \pm 0.42$ |

**(b) 5-shot mixed ways**

| | MiniImageNet | | | | FC100 | | | |
| --- | --- | --- | --- | --- | --- | --- | --- | --- |
| | 4-way | 5-way | 7-way | 10-way | 4-way | 5-way | 7-way | 10-way |
| ANIL | $70.89 \pm 0.60$ | $66.17 \pm 0.54$ | $58.24 \pm 0.43$ | $50.16 \pm 0.34$ | $52.43 \pm 0.63$ | $46.14 \pm 0.54$ | $37.84 \pm 0.44$ | $30.75 \pm 0.29$ |
| BOIL | $39.52 \pm 0.50$ | $33.47 \pm 0.42$ | $26.23 \pm 0.31$ | $20.02 \pm 0.25$ | $35.81 \pm 0.49$ | $30.30 \pm 0.41$ | $23.61 \pm 0.32$ | $17.96 \pm 0.24$ |
| MAML | $68.48 \pm 0.62$ | $63.15 \pm 0.55$ | $55.03 \pm 0.43$ | $46.62 \pm 0.33$ | $51.04 \pm 0.64$ | $44.75 \pm 0.54$ | $36.41 \pm 0.43$ | $29.60 \pm 0.30$ |
| NP-MAML (0) | $70.70 \pm 0.60$ | $65.49 \pm 0.54$ | $57.95 \pm 0.42$ | $50.47 \pm 0.36$ | $\mathbf{57.89 \pm 0.63}$ | $\mathbf{51.89 \pm 0.56}$ | $\mathbf{43.66 \pm 0.45}$ | $\mathbf{36.21 \pm 0.32}$ |
| NP-MAML (1) | $\mathbf{72.27 \pm 0.60}$ | $\mathbf{67.39 \pm 0.53}$ | $\mathbf{59.59 \pm 0.43}$ | $\mathbf{51.69 \pm 0.35}$ | $57.37 \pm 0.60$ | $50.86 \pm 0.54$ | $42.35 \pm 0.44$ | $34.58 \pm 0.31$ |
| Proto-Net | $71.66 \pm 0.73$ | $66.17 \pm 0.67$ | $58.49 \pm 0.55$ | $50.92 \pm 0.41$ | $48.55 \pm 0.82$ | $42.78 \pm 0.70$ | $35.23 \pm 0.54$ | $28.57 \pm 0.39$ |
| UnicornMAML | $68.66 \pm 0.89$ | $63.44 \pm 0.76$ | $55.48 \pm 0.62$ | $48.00 \pm 0.46$ | $51.63 \pm 0.92$ | $45.15 \pm 0.76$ | $37.65 \pm 0.60$ | $30.59 \pm 0.43$ |

Table 2: The **classification accuracy** [%] of the models on 1-shot and 5-shot in-domain for in-domain (MiniImageNet) and cross-domain (FC100) tasks, where the number of ways varies during test time. All models have been meta-trained on MiniImageNet.

In Table 2a, the results for 1-shot in- and cross-domain can be found. In the case of an in-domain adaptation, NP-MAML (fine-tuned) performs the best; however, it is only slightly better than ANIL (the difference is smaller than $1\%$). This holds true across all varying numbers of ways, which does not seem to have an impact on the performance difference. NP-MAML is, without fine-tuning, significantly worse than ANIL or MAML, but with one fine-tuning step, its performance is improved by $\approx 5\%$. BOIL performs comparatively poorly in the mixed-way setting, which confirms our claim from Section 4 that BOIL is reliant on the train-time head, reinforcing one of the motivations for our nonparametric approach. UnicornMAML performs similarly to ANIL and MAML in the in-domain tasks. Here, its trainable linear layer does not yield any improvement over NIL testing. However, in the case of cross-domain tasks, UnicornMAML is significantly better than ANIL, BOIL, and MAML. Nevertheless, it still falls short in comparison to our novel technique, NP-MAML.

In the case of a cross-domain adaptation, NP-MAML again performs best, followed by MAML with a difference of around $3.5\%$ across ways. NP-MAML without fine-tuning is only slightly worse than MAML. We observe similar results in the case of 5-shots and on more few-shot learning datasets (Appendix E).

For 5-shot learning (Table 2b), we see similar results, with the difference that for NP-MAML the fine-tuning step cannot improve performance on FC100. However, in the results reported in Appendix E, we see an increased necessity for fine-tuning NP-MAML in the case of fine-grained problems such as CUB. In addition, it seems that for the 5-shot problems in general, Unicorn-MAML is less suited than MAML or ANIL, presumably because the NIL-predictor improves more strongly with the number of available shots.

The above results show that in the cross-domain setting, the ability to fine-tune the nonparametric predictor, as is the case for NP-MAML, gains importance with a decrease in available support set samples.

### 5.3 REPRESENTATION SIMILARITY ANALYSIS

In our last experiment, we address the question *rapid learning vs. feature reuse?*, as posed by Raghu et al. (2020), for NP-MAML. Similar to Raghu et al. (2020); Oh et al. (2021); Goerttler & Obermayer (2021), we measure the centered kernel alignment (CKA) similarity of representations of the query set before and after adaptation for the three components $g_{\text{early}}$, $g_{\text{penult}}$ and $g_{\text{head}}$ of conv4 of all models we compared.

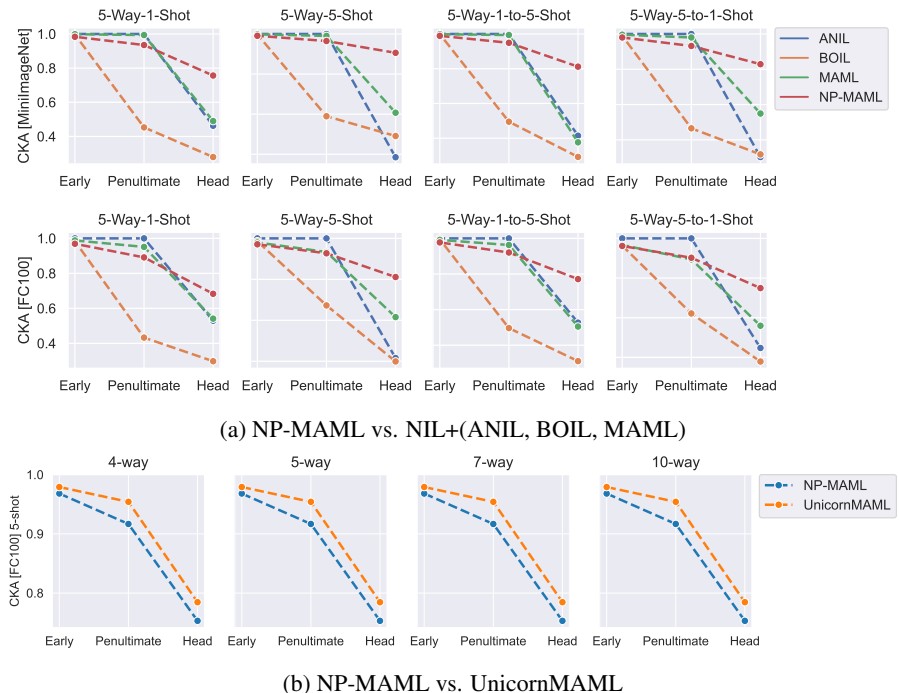

Figure 1: **CKA similarity** of network representations before and after adaptation for the models of different scenarios from previous experiments. Representation similarity was obtained for the three components $g_{\mathbf{early}}$, $g_{\mathbf{penult}}$ and $g_{\mathbf{head}}$ of the conv4 architecture.

In Figure 1a, we see the CKA similarities for different scenarios of our experiments. We can generally observe that the representation of the head changes much less for NP-MAML than for the others, meaning a good performance does not require a huge change there. In the case of cross-domain tasks, the adaption for MAML and NP-MAML is larger than in the case of in-domain tasks, underlying the overall need for representation change in cross-domain environments. For the penultimate layer, the similarity of BOIL is much less than for NP-MAML. Interestingly, all observations do not change significantly when the number of shots changes during training and testing.

We additionally analyzed how the representation change differs between UnicornMAML and NP-MAML in the setup of the mixed ways because they are the only ones who can update the weights during fine-tuning (see Figure 1b. We see that NP-MAML changes more in the early and the penultimate layers, which, as we know from the analysis done in Oh et al. (2021), can improve cross-domain adaptation capability. We hypothesize that the increased representation change of NP-MAML compared to UnicornMAML contributes to cross-domain performance in our experimental setting.

# 6    DISCUSSION

In this section, we discuss all major components of MAML during training and testing under domain adaptation: training and testing (data processing, importance of low-level features and high-level features, type of predictor).

**Task Distribution**    The results of Section 5 show that ANIL and MAML perform very well on in-domain tasks, even if the number of shots and ways is different in testing[3]. However, in the case of cross-domain adaption, we see that these algorithms perform poorly, as their features are not changing to accommodate for the domain shift. BOIL, which is good in cross-domain adaptation, fails in the case of more realistic conditions, such as a flexible number of ways, as it seems to rely

---

[3]In particular, see 5.2 and Appendix F

strongly on the head applied during training, as we showed in Sub-section 5.2. NP-MAML solves this issue by maintaining the strengths of BOIL (enforcing representation change) but still shows strong performance with flexible numbers of ways and shots.

**Fine-Tuning** As discussed in Section 3, fine-tuning on heterogeneous test environments is challenging for existing architectures, a gap filled by our approach, NP-MAML. From the results in Appendix F (Table 6b), we see that fine-tuning is not improving the results when increasing the number of shots, which we believe is partially attributed to the class prototypes that average across five samples. Still, we see a benefit to fine-tuning again in the 5-shot setting of coarse- to fine-grain domain shift, as demonstrated on the CUB dataset in Appendix E.

In addition, we observed that NP-MAML reaches their cross-domain performance with just one fine-tuning step (similar to BOIL), whereas ANIL and MAML use 5 and 10 for training and testing, respectively. We think that NP-MAML has many similarities to BOIL, e.g., strong cross-domain adaption and body-only inner learning.

We note that due to the absence of a parametric head in NP-MAML, any fine-tuning thereof is naturally omitted. However, another perspective is to view NP-MAML as employing a *pseudo-head*, a classic linear layer whose parameters are, however, implicitly formed by the output of the network body (Snell et al., 2017). We want to present this perspective to highlight that NP-MAML is not necessarily a variant of BOIL but follows a distinct paradigm of abandoning the parametric head as a standard component of meta-learning architectures.

**Early Layers** The analysis of the importance of early layers shows that penultimate-only models perform at least as well as full models in in-domain tasks but are significantly worse for cross-domain adaption (see Appendix G). The effect of only updating the penultimate layer harms BOIL more than NP-MAML, which indicates a higher resilience of our approach to this effect. Since for NP-MAML, the penultimate version is always better for in-domain tasks, it can be summarized that fine-tuning them is beneficial for cross-domain tasks, but for in-domain tasks, we better leave them not-updated. Especially in the case of larger models (e.g., ResNet), this is important to note, as it speeds up the fine-tuning quite significantly (as also argued in (Oh et al., 2021)).

**Penultimate Layer** Oh et al. (2021) have already discovered the importance of fine-tuning the penultimate layer for few-shot learning, especially for cross-domain adaptation. Our results from Section 5.1 and Appendix F confirm that this also extends to our heterogeneous setting, further strengthening the case for methods such as ours that enable strong representation change in the network body rather than the head.

**Predictor (Head)** The choice of the predictor (head) influences the generalization performance the most. In our experiments, we could show the superiority of nonparametric heads. Especially for heterogeneous tasks, it allows for a good adaptation. In addition, NP-MAML is the only model where training and test conditions match precisely. That this is desirable has been discussed by Vinyals et al. (2016).

## 7   CONCLUSION

In this paper, we proposed NP-MAML, a nonparametric version of MAML, which can deal with architectural and algorithmic challenges in few-shot learning. In addition, we explored the key components of MAML-type models and proposed a common framework to describe them. Our experimental results show that NP-MAML outperforms established methods with parametric heads. Under domain shift, our method beats ANIL and MAML due to its ability to perform representation change also in the earlier layers. In the case of heterogeneous task distributions, NP-MAML is very flexible in fine-tuning the support set, something which BOIL is limited at architecturally. In our in-depth analysis of MAML key components, we revealed the hitherto unknown importance of fine-tuning very early layers for improved cross-domain capabilities. Our representation similarity analysis showed that NP-MAML adapts less in the network body during the inner loop than BOIL; however, we demonstrate that although NP-MAML adapts its parameters less in total, the distribution of adaptation over the network components is much more uniform, which seems to be the crucial factor for performance in the presented heterogeneous tasks.

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

| Model | Configuration $\mathbf{g}^*$ |
|---|---|
| MAML (Finn et al., 2017) | $\{g_{\text{early}}, g_{\text{penult}}, g_{\text{head}}\}$ |
| ANIL (Raghu et al., 2020) | $\{g_{\text{head}}\}$ |
| BOIL (Oh et al., 2021) | $\{g_{\text{early}}, g_{\text{penult}}\}$ |
| BOIL-4 (Oh et al., 2021) | $\{g_{\text{penult}}\}$ |

Table 3: Meta-learning configurations corresponding to methods from (Finn et al., 2017; Raghu et al., 2020; Oh et al., 2021). BOIL-4 indicates a variant of BOIL where only the penultimate layer of conv4 is meta-learned.

## A    Instantiation of previous methods of our modularization

Our modularization decomposes the network into different parts Previously proposed methods (Finn et al., 2017; Raghu et al., 2020; Oh et al., 2021) can be obtained by only optimizing the weights of selected sub-components (see Table 3).

## B    (Cross-)Domain Adaptation and Generalization

The problem of domain adaptation refers to the situation where (labeled) data in one domain $A$ exists plentiful, while (labeled) data in a different domain $B$ (typically called the *target domain*) is scarce, requiring models to transfer knowledge gained in the domain $A$ to make predictions in domain $B$ (Tseng et al., 2020). As stated by III & Marcu (2006), the idea is not to ignore data from domain $B$ entirely but rather to reduce the acquisition effort by transferring from domain $A$. This is opposed to *domain generalization*, where models have no access to any data from $B$ during training. The latter is sometimes also referred to as *cross-domain transfer* (Triantafillou et al., 2020) or *cross-domain adaptation* (Oh et al., 2021).

In this work, we adopt the use of *cross-domain adaptation* as referring to both having little or no access to data from domain $B$, to stay consistent with the current literature in meta-learning (Triantafillou et al., 2020; Cai & Shen, 2020; Tseng et al., 2020).

### B.1    Multi-Modal Meta-Learning

To adapt the above distinction for another knowledge transfer problem setting, in *multi-modal learning*, one presents a model with data from both $A$ and $B$, hoping that predictive performance improves via knowledge transfer between the domains, that is, for example, reuse of general image features that can be found in both domains (Vuorio et al., 2019; Abdollahzadeh et al., 2021). Yet, knowledge transfer is not limited to such rather obvious commonalities. Vuorio et al. (2019) explicitly model such multi-modal settings with MAML and show performance improvement over a MAML model that treats each task coming from the same domain. Research in this direction has direct implications on mixtures-of-domain datasets, such as Meta-Dataset (Triantafillou et al., 2020).

### B.2    Coarse- vs. Fine-Grained Domains

Another important notion in cross-domain adaptation is the notion of *coarse-* and *fine-grained domains* (Triantafillou et al., 2020; Oh et al., 2021), referring to the level of abstractions of the "topics" contained within the data. For example, in the context of image classification, a domain containing classes like dogs, cats, elephants, and giraffes is considered coarser than a domain with classes like German Sheppard, Corgi, Husky, and Bulldog. As "level of abstraction" is not clearly defined, this notion is typically used only in distinguishing two domains, for example, when performing domain adaptation from a coarse- to a fine-grained domain.

## C    Investigating the Effect of the Nonparametric Predictor

In Figure 2, we investigate the effect of the nonparametric predictor on the features of the network body visually.

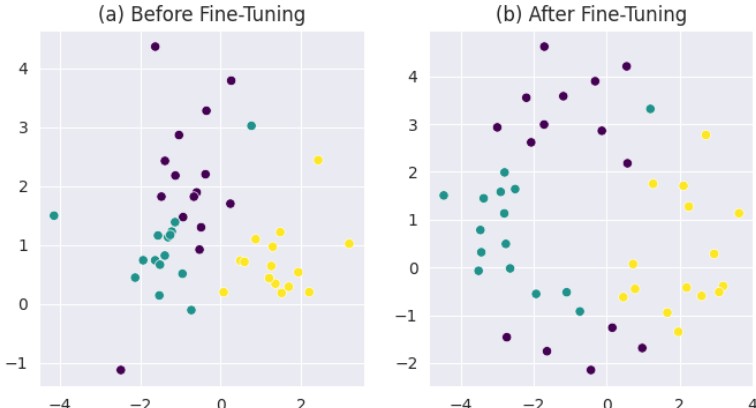

Figure 2: T-SNE plot of the features of a 3-way-1-shot task, produced by a network body trained with an NP-predictor. Features are computed from test samples of a MiniImageNet task, (a) before and (b) after fine-tuning and colored according to their ground-truth label. We observe that the NP-based fine-tuning, as introduced in Equation 12, resolves the cluttered features in the middle of the left plot, pulling features that do not belong to the same tasks apart. The model that produced the above example was trained for 30.000 iterations on 5-way-1-shot MiniImageNet, with the same hyperparameters as for our experiments in Section 5.

## D  IMPLEMENTATION DETAILS

In this section, we outline the exact implementation details of each of the models we used throughout the experiments in Section 5.

All models were trained and tested on the conv4 architecture (Finn et al., 2017; Raghu et al., 2020; Oh et al., 2021), comprised of four convolutional blocks, each containing a $3 \times 3$ convolutional layer with one stride, a 2D Batch-Normalization layer, a ReLU non-linearity and max-pooling layer with two strides. For models with parametric heads (ANIL, MAML, and BOIL), we formed predictions from the convolutional features with a linear, fully-connected layer. UnicornMAML was trained and tested always with its linear head. In its original paper, they always used pre-trained weights wherefore our results do not match their high performance.

MiniImageNet, FC100, CIFAR-FS and CUB data has been preprocessed with the data-loaders of the torchmeta library (Deleu et al., 2019). Constructing the conv4 architecture, as well as fine-tuning the models, has been aided by the learn2learn framework (Arnold et al., 2020).

### D.1  MODEL CONFIGURATIONS AND HYPERPARAMETERS

We carefully selected model configurations and their corresponding hyperparameters based on best practices in previous work (Finn et al., 2017; Raghu et al., 2020; Oh et al., 2021). In addition, to run MAML training at a similar time to the other models, we used first-order MAML throughout all of our experiments. In addition, as outlined in Section 2.2.2, we used an approximation to the meta-gradient of NP-MAML and NP-MAML-4 models. We trained and evaluated each model on three different seeds and reported averages across seeds.

Gradient-based meta-learning typically requires the following hyperparameters: A number of training iterations $T_{\textbf{train}}$, a number of test iterations $T_{\textbf{test}}$, a number of fine-tuning steps during training $s_{\textbf{train}}$, a number of fine-tuning steps during test time $s_{\textbf{test}}$, a number of test shots (in the query set) $K_{\textbf{test}}$, an outer learning rate $\beta$, an inner learning rate $\alpha$, a batch-size of the number of tasks sampled each iteration and over which the meta-gradient is formed, a flag indicating whether to approximate the meta-gradient and a method for forming predictions if nonparametric testing is required (e.g., NIL testing (Raghu et al., 2020)).

| Method | $s_{\text{train}}$ | $s_{\text{test}}$ | $\beta$ | $\alpha$ | Approximate MetaGrad | Nonparametric Predictor |
|---|---|---|---|---|---|---|
| ANIL | 5 | 10 | 0.001 | 0.01 | No | NIL |
| MAML | 5 | 10 | 0.001 | 0.01 | Yes | NIL |
| BOIL | 1 | 1 | 0.001 | 0.5 | No | NIL |
| NP-MAML | 1 | 1 | 0.001 | 0.5 | Yes | Pairw. distance |
| UnicornMAML | 5 | 10 | 0.001 | 0.01 | Yes | - |

Table 4: **Hyperparameter settings** for ANIL, MAML, BOIL, NP-MAML and UnicornMAML, used for the experiments in Section 5.

**(a) 1-shot mixed-ways on CIFAR-FS and CUB**

| | CIFAR-FS | | | | CUB | | | |
|---|---|---|---|---|---|---|---|---|
| | 4-way | 5-way | 7-way | 10-way | 4-way | 5-way | 7-way | 10-way |
| ANIL | 43.75 ± 0.96 | 37.93 ± 0.82 | 31.04 ± 0.63 | 24.48 ± 0.47 | 44.48 ± 0.93 | 38.54 ± 0.78 | 31.13 ± 0.61 | 24.18 ± 0.44 |
| BOIL | 31.13 ± 0.63 | 26.11 ± 0.54 | 19.89 ± 0.41 | 15.06 ± 0.31 | 28.48 ± 0.53 | 23.31 ± 0.45 | 17.12 ± 0.36 | 12.45 ± 0.25 |
| MAML | 44.85 ± 0.98 | 38.54 ± 0.81 | 31.09 ± 0.65 | 24.64 ± 0.47 | 43.35 ± 0.90 | 37.63 ± 0.76 | 30.19 ± 0.57 | 23.48 ± 0.43 |
| NP-MAML (0) | 46.64 ± 0.95 | 40.69 ± 0.81 | 33.69 ± 0.66 | 24.64 ± 0.47 | 41.59 ± 0.84 | 36.36 ± 0.71 | 28.81 ± 0.55 | 22.47 ± 0.41 |
| NP-MAML (1) | **52.08 ± 1.05** | **45.77 ± 0.89** | **38.15 ± 0.72** | **31.05 ± 0.53** | **48.60 ± 0.94** | **43.02 ± 0.80** | **34.80 ± 0.62** | **27.63 ± 0.46** |
| UnicornMAML | 45.46 ± 0.99 | 39.78 ± 0.82 | 32.09 ± 0.66 | 25.72 ± 0.50 | 45.07 ± 0.89 | 40.18 ± 0.73 | 32.20 ± 0.60 | 25.05 ± 0.44 |

**(b) 5-shot mixed-ways on CIFAR-FS and CUB**

| | CIFAR-FS | | | | CUB | | | |
|---|---|---|---|---|---|---|---|---|
| | 4-way | 5-way | 7-way | 10-way | 4-way | 5-way | 7-way | 10-way |
| ANIL | 60.37 ± 0.93 | 55.01 ± 0.83 | 47.05 ± 0.65 | 40.10 ± 0.51 | 61.85 ± 0.86 | 55.55 ± 0.80 | 47.80 ± 0.65 | 40.15 ± 0.52 |
| BOIL | 42.71 ± 0.82 | 37.14 ± 0.70 | 29.56 ± 0.55 | 23.70 ± 0.40 | 33.55 ± 0.59 | 28.03 ± 0.51 | 21.44 ± 0.38 | 16.16 ± 0.29 |
| MAML | 58.63 ± 0.94 | 52.96 ± 0.83 | 44.88 ± 0.65 | 30.79 ± 0.34 | 57.30 ± 0.87 | 51.01 ± 0.78 | 42.76 ± 0.60 | 35.26 ± 0.47 |
| NP-MAML (0) | **68.69 ± 0.92** | **63.62 ± 0.84** | **56.27 ± 0.71** | **49.94 ± 0.55** | 63.49 ± 0.85 | 57.82 ± 0.76 | 50.08 ± 0.64 | 42.73 ± 0.52 |
| NP-MAML (1) | 68.47 ± 0.92 | 63.18 ± 0.82 | 55.50 ± 0.69 | 48.61 ± 0.54 | **65.69 ± 0.86** | **59.78 ± 0.78** | **51.93 ± 0.64** | **44.19 ± 0.52** |
| UnicornMAML | 59.64 ± 0.96 | 53.97 ± 0.88 | 46.54 ± 0.67 | 39.53 ± 0.50 | 61.26 ± 0.88 | 54.90 ± 0.78 | 47.35 ± 0.63 | 40.33 ± 0.48 |

Table 5: Further results of **classification accuracy** [%] of ANIL, BOIL, NP-MAML, MAML, and UnicornMAML on 1-shot and 5-shot on cross-domain tasks (CIFAR-FS and CUB) with a varying number of ways during test time. The models were pre-trained on MiniImageNet.

Following previous work (Raghu et al., 2020; Oh et al., 2021), we train each model on 30.000 iterations on MiniImageNet, and following Finn et al. (2017), we evaluate test performance on a total of 500 randomly sampled tasks from the test set of the target domain (either MiniImageNet or FC100). Additionally, across all experiments and models, we set $K_{\text{test}} = 15$. Further, for 5-shot training or testing, we set the batch-size to 2, and for 1-shot training or testing, we set the batch-size to 4. All other hyperparameters are method-specific and are presented for each of ANIL, BOIL, NP-MAML, and MAML in Table 4. Penultimate-only methods BOIL-4 and NP-MAML-4 follow the hyperparameter settings of BOIL and NP-MAML, respectively.

All experiments have been conducted on a single NVIDIA Titan Xp GPU.

# E    ADDITIONAL RESULTS FOR VARYING THE WAYS

We performed additional experiments on CIFAR-FS and CUB, which can be found in Table 5. The results confirm our observation in Section 5.2.

# F    ADDITIONAL RESULTS ON THE PERFORMANCE WHEN VARYING NUMBER OF SHOTS

We repeat the experiments from Section 5.1 and 5.2 and additionally interchange the numbers of shots while testing the models (1-shot to 5-shot and vice versa).

In Table 6a, we see the results when $K$ is constant. Especially in the case of a 1-to-5-shot problem, NP-MAML is good and outperforms when we are in a cross-domain setup. Interestingly, ANIL is very bad when we have five shots in training but only 1 in training.

**(a) Switching number of shots with fixed ways**

|  | MiniImageNet | | FC100 | |
|---|---|---|---|---|
|  | 5-to-1-shot | 1-to-5-shot | 5-to-1-shot | 1-to-5-shot |
| ANIL | $27.23 \pm 0.42$ | $61.56 \pm 0.56$ | $25.05 \pm 0.35$ | $41.71 \pm 0.56$ |
| BOIL | $\mathbf{49.20 \pm 0.63}$ | $\mathbf{63.92 \pm 0.54}$ | $35.51 \pm 0.53$ | $46.83 \pm 0.52$ |
| MAML | $43.23 \pm 0.60$ | $60.42 \pm 0.57$ | $32.37 \pm 0.52$ | $45.44 \pm 0.55$ |
| NP-MAML | $45.68 \pm 0.58$ | $63.41 \pm 0.51$ | $\mathbf{35.57 \pm 0.52}$ | $\mathbf{50.28 \pm 0.53}$ |

**(b) 1-to-5-shot with mixed ways**

|  | MiniImageNet | | | | FC100 | | | |
|---|---|---|---|---|---|---|---|---|
|  | 4-way | 5-way | 7-way | 10-way | 4-way | 5-way | 7-way | 10-way |
| ANIL | $68.74 \pm 0.61$ | $63.20 \pm 0.55$ | $54.96 \pm 0.44$ | $46.54 \pm 0.34$ | $50.38 \pm 0.63$ | $44.64 \pm 0.54$ | $36.15 \pm 0.43$ | $29.24 \pm 0.31$ |
| BOIL | $38.33 \pm 0.50$ | $32.68 \pm 0.42$ | $25.34 \pm 0.32$ | $19.45 \pm 0.23$ | $35.31 \pm 0.53$ | $29.84 \pm 0.44$ | $23.02 \pm 0.33$ | $17.59 \pm 0.24$ |
| MAML | $66.40 \pm 0.64$ | $60.81 \pm 0.56$ | $52.26 \pm 0.43$ | $44.08 \pm 0.33$ | $51.12 \pm 0.66$ | $45.20 \pm 0.55$ | $36.69 \pm 0.43$ | $29.53 \pm 0.30$ |
| NP-MAML (0) | $\mathbf{69.23 \pm 0.61}$ | $\mathbf{64.16 \pm 0.54}$ | $\mathbf{56.10 \pm 0.45}$ | $\mathbf{48.43 \pm 0.35}$ | $\mathbf{56.42 \pm 0.64}$ | $\mathbf{50.91 \pm 0.54}$ | $\mathbf{42.26 \pm 0.44}$ | $\mathbf{34.65 \pm 0.32}$ |
| NP-MAML (1) | $69.18 \pm 0.59$ | $63.41 \pm 0.51$ | $54.31 \pm 0.42$ | $45.82 \pm 0.33$ | $56.18 \pm 0.65$ | $50.28 \pm 0.53$ | $41.25 \pm 0.42$ | $33.40 \pm 0.32$ |

**(c) 5-to-1-shot with mixed ways**

|  | MiniImageNet | | | | FC100 | | | |
|---|---|---|---|---|---|---|---|---|
|  | 4-way | 5-way | 7-way | 10-way | 4-way | 5-way | 7-way | 10-way |
| ANIL | $\mathbf{55.84 \pm 0.73}$ | $\mathbf{50.47 \pm 0.64}$ | $\mathbf{42.25 \pm 0.48}$ | $\mathbf{34.22 \pm 0.36}$ | $40.83 \pm 0.63$ | $35.34 \pm 0.53$ | $27.62 \pm 0.40$ | $21.53 \pm 0.29$ |
| BOIL | $31.15 \pm 0.44$ | $25.96 \pm 0.37$ | $19.44 \pm 0.29$ | $14.57 \pm 0.21$ | $29.94 \pm 0.43$ | $24.45 \pm 0.34$ | $18.40 \pm 0.27$ | $13.52 \pm 0.19$ |
| MAML | $53.86 \pm 0.73$ | $48.34 \pm 0.63$ | $39.93 \pm 0.46$ | $31.99 \pm 0.35$ | $40.49 \pm 0.63$ | $34.26 \pm 0.53$ | $27.05 \pm 0.41$ | $20.95 \pm 0.29$ |
| NP-MAML (0) | $46.52 \pm 0.67$ | $41.18 \pm 0.59$ | $33.15 \pm 0.44$ | $26.35 \pm 0.34$ | $39.75 \pm 0.59$ | $34.11 \pm 0.53$ | $26.98 \pm 0.40$ | $21.20 \pm 0.31$ |
| NP-MAML (1) | $50.91 \pm 0.68$ | $45.68 \pm 0.58$ | $37.31 \pm 0.44$ | $30.28 \pm 0.35$ | $\mathbf{41.56 \pm 0.62}$ | $\mathbf{35.57 \pm 0.52}$ | $\mathbf{27.96 \pm 0.40}$ | $\mathbf{21.88 \pm 0.29}$ |

Table 6: In these tables, the number of shots is switched between training and testing. We see the **classification accuracy** [%] of the models on a test of MiniImageNet and FC100, all trained on MiniImageNet. In (a), the number of ways is fixed, whereas, in (b) and (c), we have a flexible number of ways. In (b), the number of shots increases during testing, and in (c) decreases.

When also the number of shots varies, BOIL is again not able to achieve good results. When increasing the number of shots during testing (see Table 6b), NP-MAML and ANIL are overall the best performers in the case of in-domain, and NP-MAML is the best in the case of cross-domain. The only difference is that fine-tuning in this heterogeneous setup is no longer beneficial for NP-MAML.

When decreasing the number of shots, NP-MAML suffers from the small numbers of samples in the case of in-domain (Table 6c). Interestingly, ANIL is very good in contrast to the fixed setup of Table 6a. We fine-tune the last layer in the fixed case, whereas we use NIL-testing in the mixed setup, which seems to influence the performance in case of varying the number of shots. In the case of cross-domain, NP-MAML outperforms the other approaches.

## G  ADDITIONAL RESULTS OF EXPERIMENTS OF THE PENULTIMATE LAYER

We want to investigate the role of the earlier layer $g_{\mathbf{early}}$, which are in the standard setup of MAML and BOIL fine-tuned. We observe the significance of updating the early layers by comparing the predictive performance of BOIL and NP-MAML with only updating the penultimate layer (suffix -4) in the inner loop versus updating all layersIn Table 7, we present the results for 5-way-few-shot and mixed-way-1-shot.

In the case of fixed-way (Table 7a), the penultimate only variant performs similarly for in-domain tasks but is significantly worse for cross-domain tasks ( $2\%$ for NP-MAML and $4\%$ for BOIL). For the mixed-way setup (Table 7b), it is similar. In in-domain problems, the penultimate-only variant is slightly better, but updating all weights in the inner loop is beneficial for the cross-domain setup. For the mixed-way 1-shot in-domain case (b), NP-MAML-4 consistently performs best across all ways after fine-tuning.However, for the in-domain case, the fine-tuned NP-MAML-4 improves only slightly over the fine-tuned NP-MAML with a performance difference of about $0.5\%$. As for all mixed-way experiments, BOIL, and by extension BOIL-4, cannot compete with the other methods, and we also observe no notable performance difference between BOIL-4 and BOIL in neither the in- nor the cross-domain case. For the mixed-way cross-domain case (c), the fine-tuned NP-MAML-4

**(a) 5-way-few-shot**

| | MiniImageNet | | FC100 | |
|---|---|---|---|---|
| | 1-shot | 5-shot | 1-shot | 5-shot |
| BOIL | $50.16 \pm 0.64$ | $65.31 \pm 0.53$ | $36.03 \pm 0.57$ | $47.83 \pm 0.52$ |
| BOIL-4 | $49.61 \pm 0.62$ | $64.15 \pm 0.53$ | $32.97 \pm 0.52$ | $43.48 \pm 0.51$ |
| NP-MAML | $49.82 \pm 0.64$ | $\mathbf{67.39 \pm 0.53}$ | $\mathbf{38.57 \pm 0.57}$ | $\mathbf{50.86 \pm 0.54}$ |
| NP-MAML-4 | $\mathbf{50.73 \pm 0.62}$ | $67.13 \pm 0.53$ | $36.12 \pm 0.55$ | $48.40 \pm 0.54$ |

**(b) mixed-way-1-shot**

| | MiniImageNet | | | | FC100 | | | |
|---|---|---|---|---|---|---|---|---|
| | 4-way | 5-way | 7-way | 10-way | 4-way | 5-way | 7-way | 10-way |
| BOIL | $30.98 \pm 0.45$ | $25.90 \pm 0.38$ | $19.17 \pm 0.28$ | $14.47 \pm 0.21$ | $29.68 \pm 0.45$ | $24.50 \pm 0.36$ | $18.31 \pm 0.28$ | $13.48 \pm 0.21$ |
| BOIL-4 | $29.67 \pm 0.40$ | $24.44 \pm 0.33$ | $17.98 \pm 0.25$ | $13.14 \pm 0.18$ | $29.08 \pm 0.41$ | $24.10 \pm 0.35$ | $17.90 \pm 0.27$ | $13.18 \pm 0.20$ |
| NP-MAML (0) | $49.18 \pm 0.70$ | $43.44 \pm 0.60$ | $35.20 \pm 0.44$ | $28.38 \pm 0.34$ | $40.24 \pm 0.64$ | $34.76 \pm 0.52$ | $27.38 \pm 0.40$ | $21.24 \pm 0.31$ |
| NP-MAML (1) | $55.50 \pm 0.75$ | $49.82 \pm 0.64$ | $41.48 \pm 0.47$ | $33.59 \pm 0.36$ | $\mathbf{44.09 \pm 0.67}$ | $\mathbf{38.57 \pm 0.57}$ | $\mathbf{30.37 \pm 0.43}$ | $\mathbf{23.83 \pm 0.30}$ |
| NP-MAML-4 (0) | $49.26 \pm 0.70$ | $43.11 \pm 0.58$ | $34.74 \pm 0.45$ | $27.69 \pm 0.34$ | $37.33 \pm 0.57$ | $31.80 \pm 0.49$ | $24.63 \pm 0.40$ | $19.22 \pm 0.29$ |
| NP-MAML-4 (1) | $\mathbf{56.02 \pm 0.75}$ | $\mathbf{50.73 \pm 0.62}$ | $\mathbf{41.98 \pm 0.48}$ | $\mathbf{34.06 \pm 0.37}$ | $41.74 \pm 0.65$ | $36.12 \pm 0.55$ | $28.44 \pm 0.43$ | $22.45 \pm 0.30$ |

**(c) mixed-way-5-shot**

| | MiniImageNet | | | | FC100 | | | |
|---|---|---|---|---|---|---|---|---|
| | 4-way | 5-way | 7-way | 10-way | 4-way | 5-way | 7-way | 10-way |
| BOIL | $39.52 \pm 0.50$ | $33.47 \pm 0.42$ | $26.23 \pm 0.31$ | $20.02 \pm 0.25$ | $35.81 \pm 0.49$ | $30.30 \pm 0.41$ | $23.61 \pm 0.32$ | $17.96 \pm 0.24$ |
| BOIL-4 | $35.97 \pm 0.44$ | $30.20 \pm 0.37$ | $23.08 \pm 0.29$ | $17.54 \pm 0.22$ | $35.10 \pm 0.49$ | $29.73 \pm 0.42$ | $23.01 \pm 0.32$ | $17.49 \pm 0.23$ |
| NP-MAML (0) | $70.70 \pm 0.60$ | $65.49 \pm 0.54$ | $57.95 \pm 0.42$ | $50.47 \pm 0.36$ | $\mathbf{57.89 \pm 0.63}$ | $\mathbf{51.89 \pm 0.56}$ | $\mathbf{43.66 \pm 0.45}$ | $\mathbf{36.21 \pm 0.32}$ |
| NP-MAML (1) | $\mathbf{72.27 \pm 0.60}$ | $\mathbf{67.39 \pm 0.53}$ | $59.59 \pm 0.43$ | $51.69 \pm 0.35$ | $57.37 \pm 0.60$ | $50.86 \pm 0.54$ | $42.35 \pm 0.44$ | $34.58 \pm 0.31$ |
| NP-MAML-4 (0) | $70.07 \pm 0.60$ | $65.09 \pm 0.54$ | $57.51 \pm 0.43$ | $49.78 \pm 0.35$ | $53.23 \pm 0.60$ | $47.08 \pm 0.53$ | $38.80 \pm 0.42$ | $32.15 \pm 0.30$ |
| NP-MAML-4 (1) | $72.11 \pm 0.58$ | $67.13 \pm 0.53$ | $\mathbf{59.74 \pm 0.43}$ | $\mathbf{51.88 \pm 0.35}$ | $54.54 \pm 0.60$ | $48.40 \pm 0.54$ | $40.06 \pm 0.42$ | $33.24 \pm 0.30$ |

Table 7: **Classification accuracy** [%] of BOIL, BOIL-4, NP-MAML-4 and NP-MAML, each trained on 5-way-1-shot MiniImageNet and evaluated on both fixed-way (a) and mixed-way (b, c) problems. Since only NP-MAML and NP-MAML-4 can be fine-tuned in the mixed-way setting, NP-MAML and NP-MAML-4 results are suffixed with the number of fine-tuning steps in parentheses.

falls behind the fine-tuned NP-MAML by about $3\%$. In all mixed-way experiments, fine-tuning improves the performance of NP-MAML. The results for 5-shot are similar.

We also present comparisons of BOIL and NP-MAML with their penultimate-only counterparts for varying the number of shots between training and test time as in Section Tables 8, 9 and 10.

| | MiniImageNet | | FC100 | |
|---|---|---|---|---|
| | 5-to-1-shot | 1-to-5-shot | 5-to-1-shot | 1-to-5-shot |
| BOIL | $\mathbf{49.20 \pm 0.63}$ | $63.92 \pm 0.54$ | $35.51 \pm 0.53$ | $46.83 \pm 0.52$ |
| BOIL-4 | $48.66 \pm 0.63$ | $62.65 \pm 0.55$ | $33.08 \pm 0.50$ | $41.87 \pm 0.49$ |
| NP-MAML (1) | $45.68 \pm 0.58$ | $63.41 \pm 0.51$ | $\mathbf{35.57 \pm 0.52}$ | $\mathbf{50.28 \pm 0.53}$ |
| NP-MAML-4 (1) | $44.38 \pm 0.60$ | $\mathbf{64.10 \pm 0.51}$ | $32.60 \pm 0.48$ | $47.39 \pm 0.53$ |

Table 8: **Classification accuracy** [%] of BOIL and NP-MAML, as well as their penultimate-only counterparts BOIL-4 and NP-MAML-4. Evaluation results for 1-shot are obtained from corresponding models trained on 5-way-5-shot MiniImageNet, and evaluation results for 5-shot are obtained from corresponding models trained on 5-way-1-shot MiniImageNet. The (1) after the NP-MAML and NP-MAML-4 results indicate that one fine-tuning step was performed before test performance was measured. Improvements of NP-MAML over its penultimate-only counterpart under domain shift become more evident when either in- or decreasing shots during test time. When switching from 5 to 1 shots in-domain (on MiniImageNet), BOIL and BOIL-4 are much stronger than the -NP variants. This effect is mirrored in favor of the -NP variants when switching from 1 to 5 shots cross-domain (on FC100).

| | MiniImageNet | | | | FC100 | | | |
|---|---|---|---|---|---|---|---|---|
| | 4-way | 5-way | 7-way | 10-way | 4-way | 5-way | 7-way | 10-way |
| BOIL | 38.33 ± 0.50 | 32.68 ± 0.42 | 25.34 ± 0.32 | 19.45 ± 0.23 | 35.31 ± 0.53 | 29.84 ± 0.44 | 23.02 ± 0.33 | 17.59 ± 0.24 |
| BOIL-4 | 35.70 ± 0.44 | 29.87 ± 0.38 | 22.53 ± 0.28 | 16.91 ± 0.20 | 34.15 ± 0.49 | 28.86 ± 0.42 | 21.92 ± 0.31 | 16.66 ± 0.23 |
| NP-MAML (0) | 69.23 ± 0.61 | 64.16 ± 0.54 | 56.10 ± 0.45 | **48.43 ± 0.35** | **56.42 ± 0.64** | **50.91 ± 0.54** | **42.26 ± 0.44** | **34.65 ± 0.32** |
| NP-MAML (1) | 69.18 ± 0.59 | 63.41 ± 0.51 | 54.31 ± 0.42 | 45.82 ± 0.33 | 56.18 ± 0.65 | 50.28 ± 0.53 | 41.25 ± 0.42 | 33.40 ± 0.32 |
| NP-MAML-4 (0) | **69.53 ± 0.60** | **64.64 ± 0.53** | **56.58 ± 0.43** | 48.39 ± 0.34 | 53.29 ± 0.64 | 47.56 ± 0.53 | 39.14 ± 0.44 | 31.92 ± 0.31 |
| NP-MAML-4 (1) | 69.37 ± 0.59 | 64.10 ± 0.51 | 55.60 ± 0.41 | 47.05 ± 0.32 | 53.05 ± 0.63 | 47.39 ± 0.53 | 38.89 ± 0.43 | 31.70 ± 0.31 |

Table 9: **Classification accuracy** [%] of BOIL and NP-MAML, as well as their penultimate-only counterparts BOIL-4 and NP-MAML-4, each trained on 5-way-1-shot MiniImageNet tasks and evaluated on a varying number of ways and five shots on in-domain (MiniImageNet) and cross-domain (FC100) tasks. Since only NP-MAML and NP-MAML-4 can be fine-tuned in the mixed-way setting, NP-MAML and NP-MAML-4 results are suffixed with the number of fine-tuning steps in parentheses. In-domain results for all -NP variants differ only slightly from each other. Cross-domain results show improvements in favor of NP-MAML, highlighting the importance of early layer fine-tuning in cross-domain settings, as discussed in Section 6. BOIL-4 performance is generally worse compared to BOIL, independent of in- or cross-domain.

| | MiniImageNet | | | | FC100 | | | |
|---|---|---|---|---|---|---|---|---|
| | 4-way | 5-way | 7-way | 10-way | 4-way | 5-way | 7-way | 10-way |
| BOIL | 31.15 ± 0.44 | 25.96 ± 0.37 | 19.44 ± 0.29 | 14.57 ± 0.21 | 29.94 ± 0.43 | 24.45 ± 0.34 | 18.40 ± 0.27 | 13.52 ± 0.19 |
| BOIL-4 | 29.64 ± 0.39 | 24.23 ± 0.33 | 17.93 ± 0.25 | 13.31 ± 0.18 | 29.52 ± 0.42 | 24.19 ± 0.35 | 18.07 ± 0.27 | 13.37 ± 0.20 |
| NP-MAML (0) | 46.52 ± 0.67 | 41.18 ± 0.59 | 33.15 ± 0.44 | 26.35 ± 0.34 | 39.75 ± 0.59 | 34.11 ± 0.53 | 26.98 ± 0.40 | 21.20 ± 0.31 |
| NP-MAML (1) | **50.91 ± 0.68** | **45.68 ± 0.58** | **37.31 ± 0.44** | **30.28 ± 0.35** | **41.56 ± 0.62** | **35.57 ± 0.52** | **27.96 ± 0.40** | **21.88 ± 0.29** |
| NP-MAML-4 (0) | 43.95 ± 0.69 | 38.04 ± 0.61 | 30.07 ± 0.46 | 23.17 ± 0.37 | 35.02 ± 0.52 | 29.44 ± 0.45 | 22.95 ± 0.35 | 17.72 ± 0.27 |
| NP-MAML-4 (1) | 49.91 ± 0.69 | 44.38 ± 0.60 | 36.30 ± 0.44 | 29.04 ± 0.35 | 38.28 ± 0.56 | 32.60 ± 0.48 | 25.81 ± 0.38 | 20.19 ± 0.28 |

Table 10: **Classification accuracy** [%] of BOIL and NP-MAML, as well as their penultimate-only counterparts BOIL-4 and NP-MAML-4, each trained on 5-way-5-shot MiniImageNet tasks and evaluated on a varying number of ways and 1 shot on in-domain (MiniImageNet) and cross-domain (FC100) tasks. Since only NP-MAML and NP-MAML-4 can be fine-tuned in the mixed-way setting, NP-MAML and NP-MAML-4 results are suffixed with the number of fine-tuning steps in parentheses. When switching from 5 shots during training to 1 shot during test time, NP-MAML (1) generally performs best with higher performance differences under domain shift.

## H   EXTENDED DISCUSSION OF MAML'S MODULES

In the following two paragraphs, we give further insight into some observations from Section 5.

### H.1   FINE-TUNING EARLY LAYERS FOR CROSS-DOMAIN ADAPTATION

In this section, we discuss the interesting finding that fine-tuning earlier layers in the conv4 substantially improves classification accuracy under domain shift. In particular, the domain shift we studied shifts from MiniImageNet to FC100, both containing fairly general image features (Russakovsky et al., 2015). Previous work on understanding convolutional networks in the context of image processing has established the view that early convolutional layers typically learn low-level features like edges and corners, whereas high-level convolutional layers learn high-level features like faces or outlines of various objects (Zeiler & Fergus, 2014). We hypothesize that, thus, fine-tuning earlier layers becomes more important the more we can expect low-level features to vary. In the case of MiniImageNet and FC100, we can observe such a variation due to the fact that FC100 images are substantially smaller ($32 \times 32$ pixels) than MiniImageNet images ($84 \times 84$ pixels). As it has become customary in cross-domain few-shot-learning, images from all domains are typically resized to a common shape to fit the model architectures (Vuorio et al., 2019; Triantafillou et al., 2020; Oh et al., 2021). The effects of such a resizing on low-level image features can be observed in Figure 3.

We further highlight that conv4 with a nonparametric predictor is flexible enough to deal with different image sizes, albeit not with a different number of channels. A study on the performance differences with and without resizing on NP-MAML and especially NP-MAML-4 is suggested as future work.

### H.2   POTENTIAL OF NONPARAMETRIC PREDICTORS IN META-LEARNING RESEARCH

Our nonparametric addition to the BOIL algorithm performs favorably compared to parametric solutions in cross-domain few-shot classification. Further, as established in Section 2.2.2, our method is flexible w.r.t. the in- and output dimensionality of a given task, making it straightforward to apply in heterogeneous environments. As a result, we encourage the use of nonparametric components, especially in meta-learning, as it enables architectures to keep up with increasing task diversity (Triantafillou et al., 2020). However, we believe that the potential of rapid learning in MAML goes far beyond the scope of image classification. Consequently, future meta-learning architectures are required to become increasingly flexible without losing the ability to rapidly learn new tasks from only a few samples. We leave future work to challenge the meta-learning system even further, posing task distributions as mixtures of, e.g., predictions on time series, audio, and speech. We believe that favorable network initialization can be learned that bridges all of those domains.

Further, we recognize the use of Gaussian Processes (Rasmussen & Williams, 2006) as another way to form nonparametric predictive distributions over few-shot tasks, which can be employed for both regression and classification. Applying Gaussian Processes to meta-learning has already received attention (Myers & Sardana, 2021).

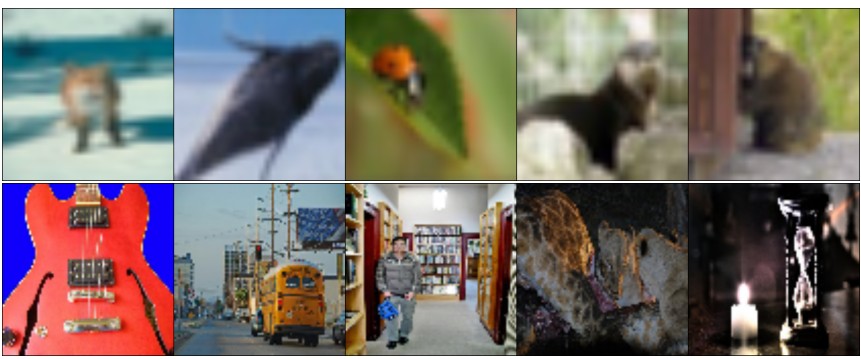

Figure 3: Image samples from FC100 (top) vs. image samples from MiniImageNet (bottom). Images have been resized to a common $512 \times 512$ pixels. We observe that FC100 samples are much blurrier when resized, requiring different features for edges, corners, etc.

