# OpenReview forum: "Towards Efficient Gradient-Based Meta-Learning in Heterogenous Environments"
_ICLR.cc/2023/Conference — Submitted to ICLR 2023_

### Official Review · Reviewer_gDT4 · 2022-10-24

**Confidence:** 3
**Correctness:** 2
**Technical Novelty And Significance:** 2
**Empirical Novelty And Significance:** 2
**Recommendation:** 6

**Clarity, Quality, Novelty And Reproducibility:**

In my opinion, this work is the combination of a prototypical network and MAML framework. There is no code for reproducibility.


**Strength And Weaknesses:**

Pros

In this work, the problem of MAML is divided into two main challenges including algorithmic challenge and architectural challenge . algorithmic challenge represents the performance of cross domain under the domain shift and architectural challenge is related to the model structure such as pixel input and the dimension of the output. These two challenges are reasonable and the results seem good in the experiment section.

Cons

NP-MAML updates the body layer with a distance using prototypes. In this respect, for the fair comparison, the architectural challenge needs to be compared with embedding methods and it is similar to embedding few-shot learning methods such as a prototypical network.

**Summary Of The Paper:**

Traditional MAML is well used in few-shot learning domains. However, recent works show that MAML does not perform well in a fast adaptation. This paper gives a detail about feature reuse phenomenon and found the relationship between lower layer and the distribution differences in training and testing.

**Summary Of The Review:**

I think this method needs to be compared to SOTA-embedding few-shot learning methods for verifying the effectiveness of MAML + prototypical network not the BOIL and ANIL which are using a fully-connected layer.

---

> ### Author Response · Authors · 2022-11-18
> **Added ProtoNet as competitor**
>
> Dear Reviewer gDT4,
>
> thanks for your comments. We came from the MAML perspective and wanted to analyze its components and their importance in the beginning and came across our analyzed weakness. Therefore, we were not considering comparing our approach to prototypical networks (ProtoNet) in the first place since these methods have already been compared in many papers.
> Nevertheless, we understood your point, think it is valid and added ProtoNet (next to UnicornMAML) as a competitor for the mixed-ways experiments. We observe that in in-domain, ProtoNet performs quite well (only slightly worse than our approach); however, if the data distribution changes, its accuracy is clearly worse than our approach, which we think is caused by no updates in the body of the layer during testing.
> We will also anonymously upload the code in the next few days to ensure the reproducibility of our results since you mentioned that they are missing.
>
> Best Regards,
> the authors

---

> > ### Comment · Reviewer_gDT4 · 2022-11-26
> > **ProtoNet**
> >
> > Thanks for your comments.
> >
> > Authors show the effectiveness of NP-MAML compared to the embedding method in data distribution changes.
> > And authors will upload the code soon. I raise my score because of the revised paper and the code issue.
> >
> > Thank you.
> > Best Regards,  Reviewer gDT4

---

### Official Review · Reviewer_nVJV · 2022-10-26

**Confidence:** 5
**Correctness:** 3
**Technical Novelty And Significance:** 3
**Empirical Novelty And Significance:** 3
**Recommendation:** 6

**Clarity, Quality, Novelty And Reproducibility:**

The work is clearly communicated and is of high quality. The novelty of the work is about 6/10, because it tackles a diverse set of under-explored task distributions, and discovers cool insights about MAML.

**Strength And Weaknesses:**

Strengths:

- Excellent and thorough empirical evaluations in terms of task formulations
- Interesting empirical insights about MAML
- Novel proposed method that is both well-motivated and effective

Weaknesses:

- The paper seems to be missing key related work, such as [1, 2, 3] and many others, but those three in particular studied MAML and came up with lots of useful insights that could have been integrated into this work.
- Possibly a side-effect of the above, the empirical experiments don't seem to incorporate some of the more powerful MAML-related models or ablation with such SOTA models along with the proposed method on top. As a result, it's hard to judge whether the proposed model is unique in its improvement in heterogeneous cases or whether its benefits disappear when used in conjunction with other powerful and existing SOTA models.

*Request: You state that there are 'no straightforward' ways of making MAML heterogeneous, however, what I've seen done, is learning a single 'class' weight matrix, that is cloned N times for an N-class problem, and then fine-tuned in the inner layer -- therefore learning a generic weight matrix that is quickly adaptable for any new N-way task coming in, no matter what N is. I'd love to see that baseline.


1. Rusu AA, Rao D, Sygnowski J, Vinyals O, Pascanu R, Osindero S, Hadsell R. Meta-learning with latent embedding optimization. arXiv preprint arXiv:1807.05960. 2018 Jul 16.
2. Antoniou A, Edwards H, Storkey A. How to train your MAML. arXiv preprint arXiv:1810.09502. 2018 Oct 22.
3. Antoniou A, Storkey AJ. Learning to learn by self-critique. Advances in Neural Information Processing Systems. 2019;32.

For additional few-shot papers that could be of interest, to ensure completeness, might be worth looking at

4. Hospedales T, Antoniou A, Micaelli P, Storkey A. Meta-learning in neural networks: A survey. IEEE transactions on pattern analysis and machine intelligence. 2021 May 11;44(9):5149-69.

**Summary Of The Paper:**

The authors investigate MAML's inner loop adaptation dynamics in a variety of different task distributions and find that in some cases, earlier layer inner loop adaptation is key for improvements, i.e. heterogeneous tasks. They also propose a method that builds on MAML that is able to improve performance in heterogeneous tasks, while remaining competitive in the remainder of the tasks.

**Summary Of The Review:**

I gave this paper a weak accept due to its interesting empirical insights and novel direction of exploring heterogeneous task distributions, and would be willing to increase my rating if:

- More SOTA models are included in the ablation studies.
- The proposed method is incorporated on top of existing and suitable SOTA methods to check if there are any benefits once a model is sufficiently strong.
- Some experiments are ran with the baseline stated in the strengths and weaknesses section, as it would dispel any doubts about how a more 'default' MAML model would perform in heterogeneous settings when it is enabled to do so by learning a quickly adaptable 'single class' weight matrix, that is cloned N times for a given N-way task and fine-tuned on the inner loop.
- More care is taken to include relevant related work, as it feels like the currently included work includes some historical work, and some more recent papers but misses much of the in-between work that is directly related with what the authors are doing in this paper.

---

> ### Author Response · Authors · 2022-11-18
> **We implemented your request as baseline**
>
> Dear Reviewer nVJV,
>
> thanks for your detailed comments on our paper. We went through the paper and added related work, including your proposed papers. Furthermore, we added more experiments and baselines.
>
> > *Request: You state that there are 'no straightforward' ways of making MAML heterogeneous, however, what I've seen done, is learning a single 'class' weight matrix, that is cloned N times for an N-class problem, and then fine-tuned in the inner layer -- therefore learning a generic weight matrix that is quickly adaptable for any new N-way task coming in, no matter what N is. I'd love to see that baseline.
>
> - We did not want to say that it is not possible, but to stress that for the original MAML and most of the extensions - having a linear classifier -  one has to change the architecture or use other techniques (like NIL). We changed our text such that this gets more clear.
> - you proposed to compare our technique with UnicornMAML (HJ Ye, WL Chao, How to Train Your MAML to Excel in Few-Shot Classification, ICLR 2022, https://openreview.net/forum?id=49h_IkpJtaE) and we think this makes total sense.
> We implemented this approach and ran all mixed-way experiments on it (Section 5.2) and also analyzed its representation (Section 5.3)
>   - UnicornMAML is in our in-domain setup in the range of ANIL, MAML, and NP-MAML
>   - in the cross-domain setup, it is better than our other MAML approaches in 1-shot and similarly good on 5-shot. However, it is significantly worse than our technique. Especially in 1-shot learning, embedding techniques have more difficulty, wherefore UnicornMAML benefits from its ability to fine-tune. In the case of multiple shots, we observe that fine-tuning is not so important anymore (also for our approach)
>   - Our representation analysis of Section 5.3 shows that it takes the benefit of having the possibility of fine-tuning earlier parts of the layer; however, similar to the original MAML in the fixed setup, it does only slightly change it. From the literature, we know that changing earlier layers is beneficial for cross-domain adaption (the reason why BOIL is so good).
>   - the motivation of UnicornMAML is to make MAML permutation invariant. Our approach is also (by design) permutation invariant, another advantage over most of the versions of MAML.
>
>
> Best Regards,
> the authors

---

### Official Review · Reviewer_wHQT · 2022-10-26

**Confidence:** 3
**Correctness:** 4
**Technical Novelty And Significance:** 3
**Empirical Novelty And Significance:** 3
**Recommendation:** 8

**Clarity, Quality, Novelty And Reproducibility:**

The work is to the best of my knowledge novel. The background, methods, experiments and discussion are all clearly written.

**Strength And Weaknesses:**

Strengths:
- Simple idea, well motivated
- Clear experiments that support the main conclusions of the paper

Weaknesses:
- Would be nice to see experiments on other domains (besides images)
- The representation similarity experiments are intriguing, but could be further developed

Minor comments:
- In the introduction, the BOIL acronym is introduced in the fourth paragraph without explaining what it is.
- Missing header in Table 1

**Summary Of The Paper:**

This paper proposes a new variant of the MAML algorithm, called "NP-MAML" for non-parametric MAML. The method uses a non-parametric head (final layer) using distances to a prototype to assign labels to new data, while still updating the body of the network in the meta-learning inner loop. The paper compares this method to related methods (BOIL/ANIL/MAML) on few-shot image classification tasks.

**Summary Of The Review:**

Overall, I thought this was a nice and interesting modification to MAML. I think the paper could be improved by demonstrating the performance of NP-MAML on additional tasks, as well as providing more intuition for what types of changes are happening in the representation (extending the CKA analysis—although perhaps this latter point is better left for future work).

---

> ### Author Response · Authors · 2022-11-18
> **We addressed your major concerns and also took your minor comments into account**
>
> Dear Reviewer wHQT,
>
> thanks for your feedback.
> In our revision, we added several experiments to further demonstrate the performance of NP-MAML on additional tasks. In addition, we extended the CKA analysis and compared the representation change of NP-MAML to UnicornMAML (a novel competitor added in the revision).
>
> Best Regards,
> the authors

---

### Official Review · Reviewer_SRwM · 2022-10-28

**Confidence:** 4
**Correctness:** 2
**Technical Novelty And Significance:** 3
**Empirical Novelty And Significance:** 2
**Recommendation:** 5

**Clarity, Quality, Novelty And Reproducibility:**

The key idea in the work is to the best of my understanding original, but the explanations in the paper and attention to detail in presentation are not sufficient, making it very hard to follow and appreciate.

**Strength And Weaknesses:**

## Major comments

The paper's key idea to try to make an algorithm like BOIL more flexible by allowing a non-parametric head makes sense and is generally well-motivated. The experimental results are also overall encouraging, though I am not sure why the proposed method performs poorly compared to baselines in Table 3c, in-domain.

However, the paper was hard to read and I feel that the main methodological idea was not that well explained or motivated. Here are a concrete set of improvement areas:

* There are a number of typos/lack of careful definitions in the presentation of the mathematical part of the paper. It is possible to decipher some of these, but it makes understanding the method a lot harder. For example: in eqn 9, what is $d$? What is $l$? I think $l$ is a typo for $k$, but this makes following it harder. In Equation 10 and 11, why are we subtracting $c_i$ and $c_j$? Are these meant to be commas (like in equation 9)? What are $d_{\text{support}}$ and $d_{\text{query}}$? What is $g_{\text{body}}(S;\theta)$?I don't think these are clearly defined enough, and it makes the contribution very hard to follow.
* The intuition behind the objective in Eqns 10 and 11 was not explained clearly enough. The figure in the appendix helps, but given this is a very important part of the paper, I think it needs more explanation. Why is this particular form of the objective better than something else?
* Relatedly, the paper says that the BOIL formulation cannot work well with the NIL head. I am not fully sure why this is the case, and the explanation in the paper was not clear enough -- this seems like the simplest baseline to try. Also, even if it doesn't work, it should be evaluated in the experiments as the most obvious baseline method.
* Given a major part of the study is domain shift, studying only one setting of domain shift from MiniImageNet to FC100 is in my opinion, not sufficient. I think the paper should at least present analysis on Cars, CUB, or something like that from the BOIL paper. In addition, it would be good to evaluate methods at least overall on Meta-Dataset, given it's a more realistic testbed for FSL methods.
* Unless I missed it, Table 3 doesn't get much attention in the text, and I would be curious to know more about why the method does worse in the in-domain setting here. The performance deltas in Table 3c, MiniImageNet are quite substantial.


## Minor
Some other typos/suggestions:
* BOIL could use a citation in the intro.
* In eq 8, the transpose operator and leading dimension are both using $T$. It's better to use a different letter for the dimension, or change the way the transpose is typeset.
* Details on proto-MAML are not very clear -- what is the subscript $k$?
* End of section 2.2.2: typo when defining the last convolutional block?
* Table 1 is missing a heading -- I think the FC100 heading is in the wrong place, and MiniImageNet heading is missing.



**Summary Of The Paper:**

* This paper proposes an approach to few-shot classification developing on related works such as MAML and Body Only Inner Loop (BOIL).
* In few-shot learning settings which involve domain shift between the training and evaluation settings,
models necessarily require some adaptation in their parameters in order to be able to solve the testing time tasks. This is because the representations learned by the earlier layers need to change in order to be effective on the new task.
* The BOIL method approaches this problem by forcing the inner layers of the network to adapt in the inner loop of meta-learning.
* However, the BOIL method is constrained in that one must pre-determine the number of output classes at training (and therefore at testing time, which must be equal to the number at train time) because this architectural property of the model cannot change once it is trained. This is a limiting factor when seeking to apply few shot models in practice, when testing scenarios may necessarily be different from training scenarios.
* This work proposes a method to combine a non-parametric classification head with an adaptation procedure for a network body. This allows both the lower level representations of the model adapt and also the number of classes at test time to vary.
* In evaluation, the paper standardises a setting where a 4-conv architecture (used in the original MAML paper) is trained on MiniImageNet, and then considers various evaluation paradigms. These include: testing on MiniImageNet (in-domain), testing on FC100 (out of domain), varying the number of ways/shots at evaluation time vs training time.
* The paper also conducts representational similarity analysis on the methods.
* The key empirical finding is that the proposed strategy can improve performance in settings over BOIL, MAML, and ANIL (uses no inner loop during training for the network body) where domain adaptation and changes in ways/shots between train and test time occur.
* In settings where one or both of these do not change, the method mostly performs competitively with the other strategies.



**Summary Of The Review:**

Overall, I think this idea has promise. However, there are too many errors in the paper's presentation and a general lack of motivation about the method used for it to be accepted at this stage. I think the paper needs substantial revisions and cleaning up in the methodological presentation, as well as some more comprehensive experiments in the domain-shift setting (with other datasets), before I can recommend it for acceptance.

---

> ### Author Response · Authors · 2022-11-18
> **Response to all major comments**
>
> Dear Reviewer SRwM,
>
> Thanks for your very detailed review, which precisely points out what you are missing.
> > There are a number of typos/lack of careful definitions in the presentation of the mathematical part of the paper.
>
>  We reworked the mathematical part such that it is easier to follow. Thanks for all the hints. We addressed all of them by either simplifying, correcting, or better describing them.
> > The intuition behind the objective in Eqns 10 and 11 was not explained clearly enough...
>
> We added more intuition on why former Eqns 10 and 11 (now Eqns 11 and 12) are formulated like they are by extending the explanation surrounding them.
> > Relatedly, the paper says that the BOIL formulation cannot work well with the NIL head. I am not fully sure why this is the case, and the explanation in the paper was not clear enough ...
>
> In Section 5.2, we did it but admittedly did not stress this clearly enough. In Section 5.2 (Table 2) we used NIL-Testing for all MAML variants except our approach and our new competitor UnicornMAML. We added already in Section 4 a paragraph explaining why this does not work well. BOIL cannot work well with the NIL head because BOIL is meta-trained to update earlier layers (including the penultimate one) to match the meta-trained head. With NIL testing, we do not change any of the representation of the network, wherefore the results are really bad for it (Table2)
> > Given a major part of the study is domain shift, studying only one setting of domain shift from MiniImageNet to FC100 is in my opinion, not sufficient...
>
> We also added experiments to emphasize our findings and strengthen them. We did experiments on CIFAR-FS and CUB and added two competitors (UnicornMAML and ProtoNet).
> > Unless I missed it, Table 3 doesn't get much attention in the text, and I would be curious to know more about why the method does worse in the in-domain setting here. The performance deltas in Table 3c, MiniImageNet are quite substantial.
>
> We discussed former Table 3 in former Section 5.3 however had a small typo in referencing it. This part can now be found in Appendix F (and Table 6). Our method is not good if we trained on 5 shots but only had 1 shot during training; other methods are better. We think that the reason for that is that our approach is meta-trained to fine-tune on 5 shots and 1 shot is not enough. However, in the case of a cross-domain adaptation, NP-MAML is still slightly better than ANIL.
>
> We also liked your minor comments and included them all in our revision.
>
> We hope that our revision convinces you and you feel that we addressed your concerns appropriately.
>
> Best Regards,
> the authors

---

> > ### Comment · Reviewer_SRwM · 2022-11-25
> > **Response to comment**
> >
> > Thanks for replying to my comments and updating the paper.
> >
> > I think the revised version of the paper is clearer and stronger. I appreciated the additional validation on two more cross-domain adaptation datasets, and clarity as to why BOIL does not immediately work with a non-parametric head. I am encouraged by the improvements in performance achieved by NP-MAML on the cross-domain tasks.
> >
> > I think the presentation of the method is clearer in this revision. However, I still feel that the presentation of the method could be clearer. I appreciate the additional clarity in section 4; however, I still think that that justification for the update in Eqn 12 is insufficient. Why does this particular form of inner parameter update make sense, as opposed to something else? Also, there could still be some more cleaning up of the presentation (for example, the subscript $j$ in eqn 12 -- what is this?). The presentation could also be improved with an algorithm box -- details such as a stop gradient application should be made more obvious.
> >
> > Overall, the revision encourages me to raise my score a point but I still do not feel that the paper is strong enough to merit acceptance in its current form. Further clarity on the specific nature of the inner update and why this is chosen would help. In addition, the paper would still benefit from a careful read-through to correct some typographical errors.

---

> > > ### Author Response · Authors · 2022-11-29
> > > **Providing updated formulas to increase clarity of approach**
> > >
> > > Dear reviewer SRwM,
> > >
> > > Thanks a lot for appreciating our improvements.
> > >
> > > > I think the presentation of the method is clearer in this revision. However, I still feel that the presentation of the method could be clearer. I appreciate the additional clarity in section 4; however, I still think that that justification for the update in Eqn 12 is insufficient. Why does this particular form of inner parameter update make sense, as opposed to something else?
> > >
> > > Based on our motivation, we saw two basic requirements for the loss function:
> > > - It has to be non-parametric
> > > - It must allow us to back-propagate through the fine-tuning
> > >
> > > We think any such loss function enables us to combine the benefits of non-parametric predictors, such as the NIL method, with the strong adaptation capabilities of MAML-type learning. The latter is partly due to the harmonization of training and test objective. This view is also mentioned in [1]. We believe this to be a crucial factor in designing meta-learning methods, further emphasized by our empirical results.
> > > The loss we propose is, however, not the only possible candidate satisfying the above requirements, and we strongly encourage an extended study of suitable non-parametric losses in our framework in future work, where our loss may serve as a baseline.
> > > As our fine-tuning closely resembles a self-supervised learning setting, we found it apt to choose a contrastive-type loss, disentangling intra-class from extra-class features, which is fairly common in self-supervised learning [2].
> > > We admit this perspective wasn't made entirely clear in the paper and are committed to including a paragraph similar to the above in an updated version.
> > >
> > > > Also, there could still be some more cleaning up of the presentation (for example, the subscript  in eqn 12 -- what is this?)
> > >
> > > We agree that Equation 12 is sloppy and does not express our method properly.
> > > We wanted to iterate with $i$ through all samples and with $j$ to all labels; however, that formulation is not clear.
> > > Therefore we would replace Equation 12 $U(S; \theta) = \theta - \frac{\alpha}{K} \nabla_{\theta} \Big( \sum_{i, j} -d(g_\textbf{body}(S_i; \theta),c_j) \Big)$ with
> > > \begin{equation}
> > >    U(S; \theta) = \theta - \frac{\alpha}{K} \nabla_{\theta} \Big( \sum_{(x, \cdot) \in S} \sum_{i=1}^{N} -d(g_\textbf{body}(x; \theta),c_i) \Big)
> > > \end{equation}
> > > in an updated version.
> > > The first summation iterates over all inputs $x$ of the support set $S$ and the second summation over all different labels. So we sum up all distances $d(\cdot, \cdot)$ between the hidden representation of every sample $x$ to every prototype ($c_i$, we do not use $j$ anymore). In the optimization step, we maximize the distances (by minimizing the negative) to disentangle the samples from all prototypes, even the correct prototype.
> > >
> > > Maximizing the distance to the correct prototype has - in the case of 1-shot learning - no influence since the distance to itself is always zero. In the case of $k$-shot learning with $k>5$, every sample is still part of the correct centroid, so maximizing the distance to it is naturally bounded. Nevertheless, one could also think about minimizing the distance to the correct prototype. This either will increase the performance or lead to overfitting to the support set and can be found out in further experiments. We think that our instantiation benefits from its *simplicity* without too much modification.
> > >
> > > In addition, we will change Equation 9 (the definition of prototypes) to
> > > \begin{equation}
> > >     c_i := \dfrac{1}{K} \sum_{(x, y) \in S} g_\textbf{body}(x, \theta) \cdot \mathbb{1}_{y=i}
> > > \end{equation}
> > > in order to align with the syntax of Equations 11 and 12.
> > >
> > >
> > >  > The presentation could also be improved with an algorithm box -- details such as a stop gradient application should be made more obvious.
> > >
> > > This is a great idea. We will provide a pseudo-algorithm in the next few days.
> > >
> > >
> > > Best regards,
> > > the authors
> > >
> > > [1] Oriol Vinyals, Charles Blundell, Tim Lillicrap, Koray Kavukcuoglu, Daan Wierstra:
> > > Matching Networks for One Shot Learning. NIPS 2016: 3630-3638
> > >
> > > [2] Ting Chen, Simon Kornblith, Mohammad Norouzi, Geoffrey E. Hinton:
> > > A Simple Framework for Contrastive Learning of Visual Representations. ICML 2020: 1597-1607

---

### Author Response · Authors · 2022-11-18
**Updates of our revision**



Dear Reviewers,

Thank you for your feedback on our manuscript.
In the last two weeks, we revised our paper and added several points having been addressed in your reviews.
To simplify finding the major changes, we want to point them out, such that you do not have to find them by comparing the documents.
Nevertheless, we invite you to have a look at our revision.

- **Clarity and Motivation**
Some of you mentioned that the approach could be better motivated. In addition, several small mistakes and typos inside the formulas made it difficult to follow the approach. We went through it and improved the formulas and its explanation such that it is now easier to understand. In addition, we also added more motivation to our method (e.g. by adding paragraphs in Section 4).

- **Experiments**
Some of you asked for more experimental results.
  - We applied our mixed-way experiments on two novel datasets (CIFAR-FS and CUB). We found out that four coarse-grained datasets (FC100 and CIFAR-FS) fine-tuning is less important and even endangers overfitting when having a 5-shot problem. For fine-grained datasets (CUB) however, fine-tuning is important.
  - In addition, we implemented two novel methods as competitors. Firstly, ProtoNet as a baseline embedding technique, and secondly, UnicornMAML, an approach that has a linear layer that is flexible to mixed ways. Our approach outperforms these two methods also, especially in a heterogeneous environment.
  - Furthermore, we also extended our representation similarity analysis part by comparing our technique NP-MAML with UnicormMAML. This is particularly interesting since both methods allow change of the weights in the earlier part of the network during fine-tuning also in the mixed-way setup.

- We added several papers as related work and revised the overall story.

- We divided our experiments into major experiments and ablation studies and put the ablation studies in the appendix to have more space for novel experiments, a better description of our method, and a clear presentation of the significant results.

- We have also realized that we did not highlight well enough our main contribution. While current MAML techniques can already partially deal with different ways during training and testing (e.g., by NIL testing), we found out that learning has to happen in the earlier part for cross-domain adaptation (the main result of BOIL). However, this is either impossible by design (NIL testing) or does not happen enough (UnicornMAML). We improved our paper to make that clearer and also added experiments to support our claim (Section 5.3).

Your comments, questions, and requests of your will be directly answered below your review. We are looking forward to your answers.

Best Regards, the authors

---

### Decision · Program_Chairs · 2023-01-20

**Decision:**

Reject

**Justification For Why Not Higher Score:**

unsatisfying explanation of the central part (eq. 12)

**Justification For Why Not Lower Score:**

The experimental results are good.

**Metareview: Summary, Strengths And Weaknesses:**

The idea of the paper is to build a few-shot learning model which can be seen as a combination of prototypical networks and MAML.
1) The method uses neg-distances to prototypes (similarly to PN) as logits. This allows to have a different number of classes at test time.
2) The method adjust the encoder using several steps of gradient descent (GD) similarly to MAML. This allows the model to improve its performance in cross-domain adaptation.

The central element of this idea is the loss which is minimized with GD to adapt the encoder. The authors propose to *maximize* the distance to the prototypes, which is a somewhat unintuitive choice.
The authors compare the methods to existing few-shot learning algorithms on a few benchmarks and report good results obtained with the proposed algorithm.

During the discussion, the reviewers raised the following concerns:

- The lack of clarity of the presentation. Although the clarity has been improved a lot during the discussion, the current version of the paper still contains errors and typos. The central part of the proposed method (equations 11 and 12) is not explained well and it required further corrections and clarifications on the openreview discussion page. The algorithm box that the reviewers posted on openreview was also very helpful and should be included in the paper.

- The motivation for the update rule (12) (which is the central contribution of the paper) remains unclear. The authors do not provide a satisfying explanation/intuition to why maximising the distance is a good idea. The explanation provided on openreview "we found it apt to choose a contrastive-type loss, disentangling intra-class from extra-class features, which is fairly common in self-supervised learning [2]." is not very convincing. It is unclear why it is a contrastive loss and what kind of disentanglement it encourages. The readers would benefit from seeing a discussion of other possibilities for the loss function and why the proposed loss is a good choice. For example, why not to use the standard PN loss for adapting the encoder?

- Too few benchmarks and baselined used in the experiments. The authors have addressed this concern by including more benchmarks (CIFAR-FS and CUB) and more baselines (Unicorn MAML and PN).

In conclusion, the reviewers think that this is a promising piece of work and the paper is very close to meet the acceptance criteria. However, given a relatively minor novelty, they believe that more polishing is required to fully meet the acceptance criteria. The authors are encouraged to incorporate all the clarifications requested by the reviewers and provide more intuition/motivation for the loss used in the adaptation phase.

**Summary Of Ac-Reviewer Meeting:**

During the meeting, we acknowledged that this is a promising piece of work and the results are encouraging, although the novelty is not very large. The two main concerns were emphasized during the discussion:

1) The lack of clarity in the paper and a substantial number of errors/typos in the submission. The clarity has been improved a lot during the discussion on openreview but all of the extra clarifications should be incorporated in the paper.

2) The lack of the intuition/motivation behind the central idea (equations 11 and 12). The authors do not provide a satisfying explanation/intuition to why maximising the distance in the adaptation phase is a good idea. The explanation provided on openreview "we found it apt to choose a contrastive-type loss, disentangling intra-class from extra-class features, which is fairly common in self-supervised learning [2]" is not convincing. It is unclear why it is a contrastive loss and what kind of disentanglement it encourages. The readers would benefit from seeing a discussion of other possibilities for the loss function and why the proposed loss is a good choice. For example, why not to use the standard PN loss for adapting the encoder?

In summary, the paper has been improved a lot during the discussion and it is very close to meet the acceptance criteria. However, given a relatively minor novelty, we believe that more polishing is required to fully meet the acceptance criteria.